# Tuning the interactions between chiral plasmonic films and living cells

Xueli Zhao[1,2], Liguang Xu[1,2], Maozhong Sun[1,2], Wei Ma[1,2], Xiaoling Wu[1,2], Chuanlai Xu [1,2] & Hua Kuang[1,2]

Designing chiral materials to manipulate the biological activities of cells has been an important area not only in chemistry and material science, but also in cell biology and biomedicine. Here, we introduce monolayer plasmonic chiral Au nanoparticle (NP) films modified with L- or D-penicillamine (Pen) to be developed for cell growth, differentiation, and retrieval. The monolayer films display high chiroptical activity, with circular dichroism values of 3.5 mdeg at 550 nm and 26.8 mdeg at 775 nm. The L-Pen-NP films accelerate cell proliferation, whereas the D-Pen-NP films have the opposite effect. Remote irradiation with light is chosen to noninvasively collect the cells. The results demonstrate that left circularly polarized light improves the efficiency of cell detachment up to 91.2% for L-Pen-NP films. These findings will facilitate the development of cell culture in biomedical application and help to understand natural homochirality.

[1] State Key Lab of Food Science and Technology, Jiangnan University, Wuxi, Jiangsu 214122, China. [2] International Joint Research Laboratory for Biointerface and Biodetection, School of Food Science and Technology, Jiangnan University, Wuxi, Jiangsu 214122, China. Xueli Zhao and Liguang Xu contributed equally to this work. Correspondence and requests for materials should be addressed to H.K. (email: kuangh@jiangnan.edu.cn)

As one of the most typical biochemical signatures of life, chirality[1–3] has a remarkable influence on many biological events[4–7]. For example, the high selectivity of biological systems for chiral species (L-amino acids, D-sugars, and so on) plays crucial roles in maintaining the normal functions of living cells[8, 9]. Most cell functions, such as adhesion[10–12], proliferation[13, 14], and differentiation[15–19], de-differentiation[20, 21] are affected by the interactions between the cells and the extracellular matrix (ECM)[22–30]. Consequently, designing biomaterials with distinct characteristics that mimic the native ECM for use in tissue regeneration or injury recovery has drawn increasing attention in biomedical research[31–34]. Adsorbing or grafting chiral bioactive ligands to surfaces has been explored in pioneering research, to manipulate biocompatibility, cell adhesion, and cell growth[35–37]. Bioactive supramolecular materials formed through incorporation of chiral groups into building blocks showed good potential as biocompatible scaffolds[38–40]. Ding's group previously investigated the adhesion and differentiation of stem cells on gold coating (glass sputtered with gold) modified with L- or D-cysteine[36], and Kehr and colleagues reported cell interactions with a chiral-amino-acid-functionalized zeolite nanoparticle (NP) monolayer[41]. These studies have shown that cells can interact with chiral surface molecules and display different behaviors on enantiomorphous surfaces. The cells were found to adhere on the D-surface much less than that on the L-surface[42–44]. The use of monolayer NPs instead of simple substrate surfaces allows much higher functional group densities, providing larger numbers of chiral contact points between the cells and the ECM. However, monolayer film fabricated with plasmonic NPs with strong chirality[45–49], for using as an ECM, has not yet been reported. As well as the chirality of the molecules in the ultraviolet region, the chiral NP films with highly intense circular dichroism (CD) peaks at 550 and 775 nm play a vital role in cell–ECM interactions.

Neurite outgrowth is an important step preceding the development of nerve systems[50, 51]; so much attention has been paid to the relationship between neuronal behavior and the surface topography[52, 53]. Recent research has demonstrated that nanostructured surfaces can accelerate neurite outgrowth and the polarization of neurons[54–57]. Choi and colleagues cultured neurons on a silica-bead monolayer to explore the biological relationships between environmental topographical features and the morphogenesis of neurons[58]. The same group also reported that neurons on vertical nanowires displayed axon-first neuritogenesis[59]. However, the chiral effects of bioactive agents on neurite outgrowth at the nanoscale have not yet been examined. The chirality of an Au film has very noticeable effects. Inducing directional cell differentiation and neurite outgrowth by dictating the chirality of the ECM may provide an appropriate strategy for the treatment of many neurological conditions and traumas.

The controlled and noninvasive retrieval of cells from responsive substrates is crucial in regenerative medicine and tissue engineering[60, 61]. Compared with conventional procedures, such as digestive enzymes, that might irreversibly damage the cells, irradiation with light to trigger cell detachment has captured the attention of investigators[62]. Qu's group used a spiropyran-conjugated upconversion NP as a photoswitch to control cell detachment[63]. Liz-Marzan and coworkers recently designed a plasmonic substrate for cell growth and detached the cells with near-infrared (NIR) light[64]. However, considering the heating effect of an NIR laser, its detrimental effect on cell viability cannot be ignored. A plasmonic film that possesses chirality can strongly rotate light, allowing the differential absorption of left- and right-handed light[3]. Therefore, to improve the efficacy of light in cell detachment and to introduce ECM protection for cells during the detachment process, circularly polarized light (CPL) could be a promising option.

In this text, bioactive films with high chirality are fabricated to interact with cells and collect cells noninvasively. The chiral Au NP film has deep influence on cell growth and differentiation. Noticeably, CPL (808 nm laser) is adopted to significantly improve the cell detachment efficacy and avoid cell damage.

## Results

**Monolayer chiral Au NP films design.** Here, we designed monolayer chiral Au NP films to promote in cell adhesion, growth, differentiation, and to be easily removed without damage. As illustrated in Fig. 1a, NG108-15 cells were seeded on L- or D-Pen-NP (Au NPs modified with L- or D-penicillamine) films supported by polydimethylsiloxane (PDMS), at the same initial density. The cells on the L-Pen-NP film proliferated faster than those on the D-Pen-NP film and stretched more efficiently. After the addition of retinoic acid (RA) as the inducer, the stretched cells differentiated into bipolar neurons, whereas the round cells became multipolar neurons. According to previous studies, NPs with diameters >20 nm usually deviate from a perfectly spherical shape[1]. The distinct dihedral angle between the constituent NPs contributes to the generation of the CD signal. Therefore, we chose Au NP ($47 \pm 5$ nm) as the building blocks for film preparation. A seed-mediated growth strategy[65] was used to synthesize uniform Au NPs (Supplementary Figs. 1 and 2), using trisodium citrate as the reducing agent and 13 nm Au NPs as the seeds. After the Au NPs were functionalized with L- or D-penicillamine (Pen, Supplementary Figs. 3 and 4), a large area of monolayer Au NP film was formed at the water–hexane interface (Supplementary Figs. 5 and 6a, Supporting Information). The monolayer film was transferred to PDMS with the Langmuir–Blodgett transfer technique[66] (Supplementary Fig. 6b). Transmission electron microscopy (TEM) and scanning electron microscopy images confirmed the formation of uniform monolayer films with close packing (Fig. 2a, b), and atomic force microscopy showed the vertical dimension to be $47 \pm 5$ nm (Fig. 2c, d, Supplementary Fig. 7). X-ray photoelectron spectroscopy (XPS) characterization was carried out to confirm the similar amounts of L-Pen and D-Pen grafted onto the Au NP films (Supplementary Fig. 8). The localized surface plasmon resonance was quite different after film formation. As shown in Fig. 2e, the peak at 550 nm shifted from the single peak at 530 nm (Supplementary Fig. 3), and a broad band at 700–900 nm emerged, providing the opportunity to use NIR light for cell detachment. This broad plasmon band might originate from extensive plasmon coupling at the small interparticle distance. The CD signals (Fig. 2f) of the L-Pen-NP and D-Pen-NP films were equal in intensity but opposite in sign. The CD intensity varied with the change of angle between the film and the light, and was maximum when the angle was rotated to 45° (Supplementary Fig. 9). As a control, PDMS sputtered with gold (Au coating) using a sputter coater displayed no CD signal (Supplementary Figs. 10 and 11). However, the NP film acquired with no chiral Pen modification showed a CD value of 1.4 at 775 nm (Fig. 2f). After coupling with L- or D-Pen, a CD peak with a value of 0.23 at 630 nm appeared in the spectrum of the Au coating (Supplementary Fig. 12). However, when the NP film was functionalized with Pen after film formation, the CD value at 775 nm was 7.8 (Supplementary Fig. 13), which is much higher than that for the Au coating but lower than that for the L- or D-Pen-NP film. Therefore, plasmonic chirogenesis may be ascribed to the following three factors: (1) Au NPs ($47 \pm 5$ nm) with an aspect ratio of $1.4 \pm 0.2$ (Supplementary Fig. 14) appear to be ellipsoids rather than spheres, so the balance between electrostatic repulsion and Van

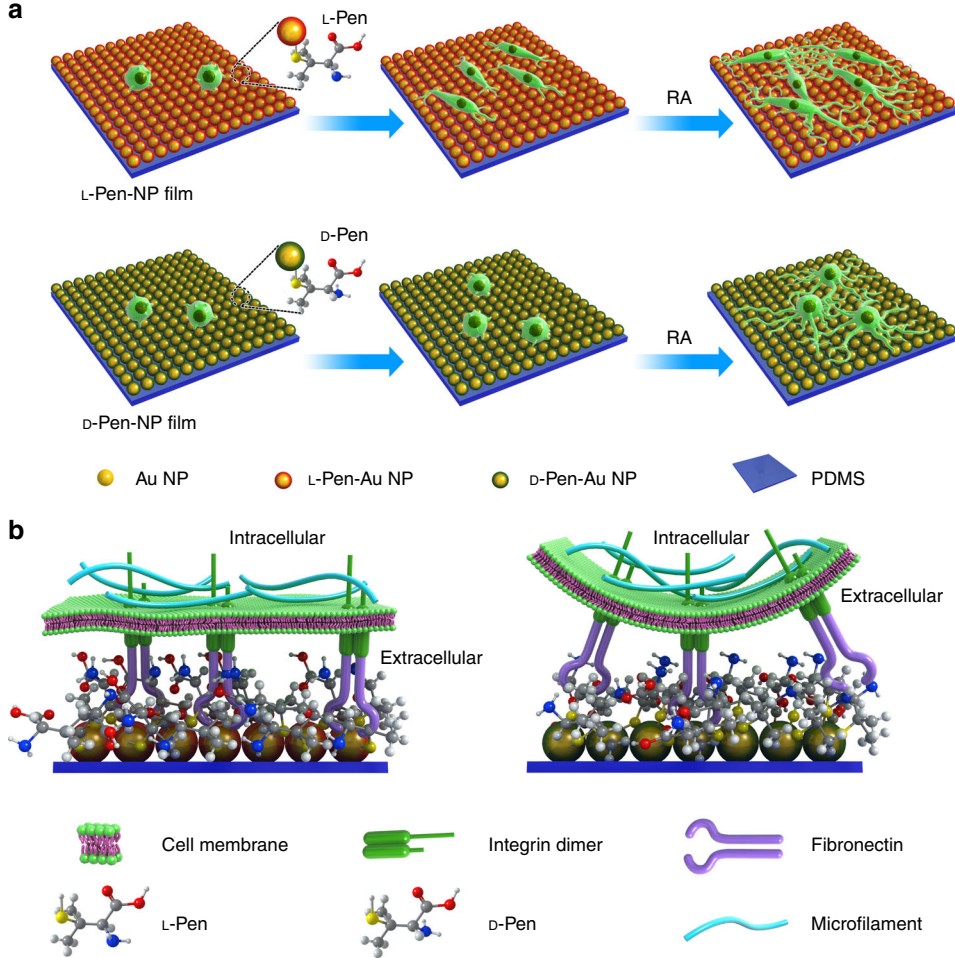

**Fig. 1** Interactions between chiral NP films and NG108-15 cells. **a** Schematic representation of NG108-15 cells grown and differentiated on chiral plasmonic films. **b** Schematic presentation of the regulation of NG108-15 cell adhesion by stereospecific interaction between fibronectin and chiral NP film

der Waals forces contributes to the dihedral angle between adjacent NPs in the monolayer film; (2) because Pen has a positive stereo-configuration, the spatial conformation of the NPs was reconfi-gured, enhancing the chiral configuration; and (3) the plasmonic hotspot enhanced the near-field dipolar Pen-gold NPs interaction[1, 67]. According to electric field simulation (Supplementary Fig. 15), with the gap of 1.5 nm, the electric field 43.6 V/m that indicates 10.7 times enhancement compared to single NP (4.06 V/m). Addition-ally, CD and ultraviolet–visible (Vis) spectra of the chiral NP films have been simulated as Supplementary Fig. 16 displayed, which was in consistent with the experimental results.

**Cells adhesion and growth on chiral substrates**. Figure 3 shows the growth of NG108-15 cells on chiral substrates. A confocal image (Fig. 3a) illustrates the cellular morphology on different surfaces. Compared with the PDMS substrate, the variation in the cellular morphology on the Au NP film was quite small. However, the state of the cells on the chiral NP films varied significantly. Most adherent cells showed stretching when grown on the L-Pen-NP film, whereas the cells on the D-Pen-NP film had a pre-dominantly round morphology. The aspect ratio of the cells on the L-Pen-NP film increased markedly, reaching a value as high as 5.3 ± 1.2, which was ~3.3 times higher than that on the D-Pen-NP film (Fig. 3b). With the same seeding density of cells on the NP film, L- and D-Pen-NP films, and PDMS surface, different cell proliferation behaviors were observed. After 48 h, the quantity of

cells on the L-Pen-NP film was 2.2 times higher than that on the D-Pen-NP film and 1.8 higher than that on the NP film (Fig. 3c, Supplementary Fig. 17). However, these differences were quite small when the cells were incubated on Pen-functionalized PDMS or Au coating compared with the differences between cells cul-tured on L- or D-Pen-NP films (Supplementary Figs. 18–20). These data suggest that not only the surface molecular chirality but also the molecule–substrate interaction play significant roles in cell adhesion. To further investigate the stereospecific interaction between cells and enantiomeric NP films, the adsorption behavior of fibronectin (FN) (a well-known protein that promotes cell adhesion) on experimental substrates was tested using an enzyme-linked immunosorbent assay (ELISA) kit[68]. As the time-dependent adsorption curves show (Supplementary Figs. 21 and 22), the quantity of FN adsorbed onto the L-Pen-NP film reached 200 ± 6.8 pg/cm$^2$ after 60 min, which was 1.74 times greater than that on the D-Pen-NP film and much higher than for previously reported materials (114 pg/cm$^2$)[43]. Therefore, the distinct cell morphologies and proliferation behaviors of cells on enantio-meric NP films might be attributable to the different signals released from the stereospecific interactions between FN and the chiral NP films (Fig. 1b). The higher density of FN adsorbed to the L-Pen-NP film provided more anchoring points for cell adsorption, resulting in stretched cell morphology and a high aspect ratio. Furthermore, when the viability of the cells on the L-Pen-NP film was analyzed with the Cell Counting Kit-8 (CCK-8), it exceeded 120% (Supplementary Fig. 23), which

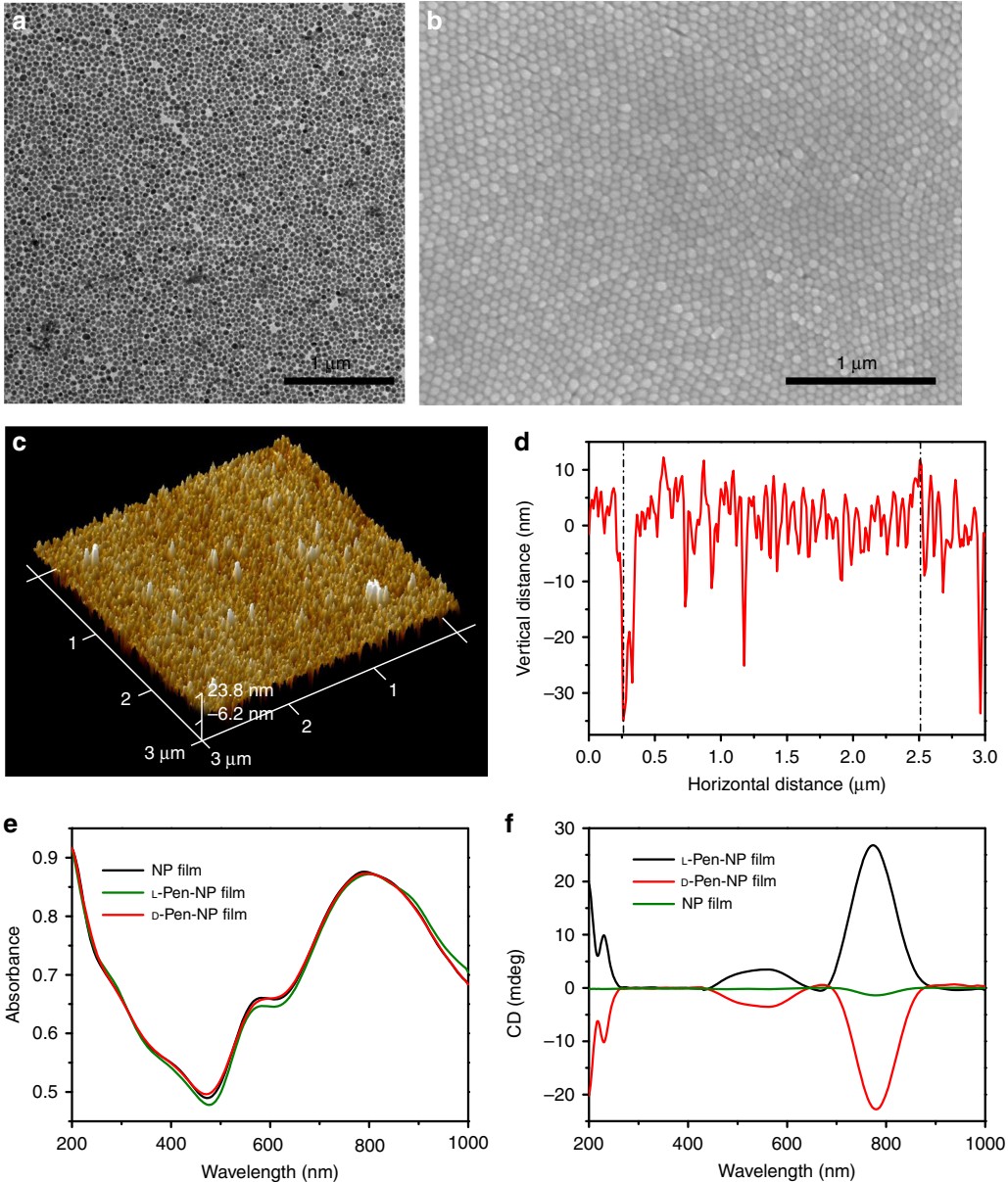

**Fig. 2** Characteristics of Au NP films. **a** TEM images of Au NP film. **b** SEM images of Au NP film. **c** AFM 3D image of Au NP film. **d** Vertical distance of Au NP film. **e** UV–Vis spectra of Au NP film and L/D-Pen modified Au NP film. **f** CD spectra of Au NP film and L/D-Pen modified Au NP film. The scale bars of TEM and SEM are 1 μm

confirmed the excellent biocompatibility of the film[36]. The CD signals of the chiral NP film were also astonishingly enhanced after cell adhesion and proliferation (Fig. 3d). A closer look at the spectrum indicated that the enhancement of the CD signal follow the same trend as the cell density change. To validate this conjecture, the cells adhering to the chiral films were treated with trypsin–EDTA solution. The resulting spectra (Fig. 3e) showed that the CD signal reverted to the original intensity in the Vis and NIR regions, confirming that the cells adhering to an NP film influence the spectral features of the film in the Vis–NIR region. The enhanced CD signals might be ascribed to the induction of the intrinsic chirality of cells[6, 69]. Cells could form invariant chiral alignment dependent on the cell phenotype and actin function, indicating that all cells are intrinsically chiral[70]. Herein, the chiral cells–plasmon dipolar interaction and plasmonic hotspot resulted in the enhancement of CD signals.

**Distinct cells differentiation on the chiral films**. Figure 4 shows the distinct differentiation states of NG108-15 cells on the chiral films. After equivalent cell seeding, RA was added to the nutrient medium as an inducer[71, 72]. As the confocal images show (Fig. 4a, Supplementary Figs. 24–28), increased neurite outgrowth of the differentiated NG108-15 cells was observed after RA treatment. The cell densities on the substrates also increased slightly (Supplementary Fig. 29). Most cells on the L-pen-NP film displayed bipolar differentiation. In contrast, the cells on the D-pen-NP film showed multipolar neurite outgrowth. The cells grown on the chiral films displayed faster neurite growth than the cells on PDMS or NP film. In particular, the cells on the L-pen-NP film showed the longest neurites, in both mean length and maximum length (Fig. 4c, d). These unique differentiation behaviors might be explained as follows. First, the Pen on the surface of the NP film can regulate multiple cellular functions[73]. Second, the stereospecific interaction between FN and the chiral NP films caused

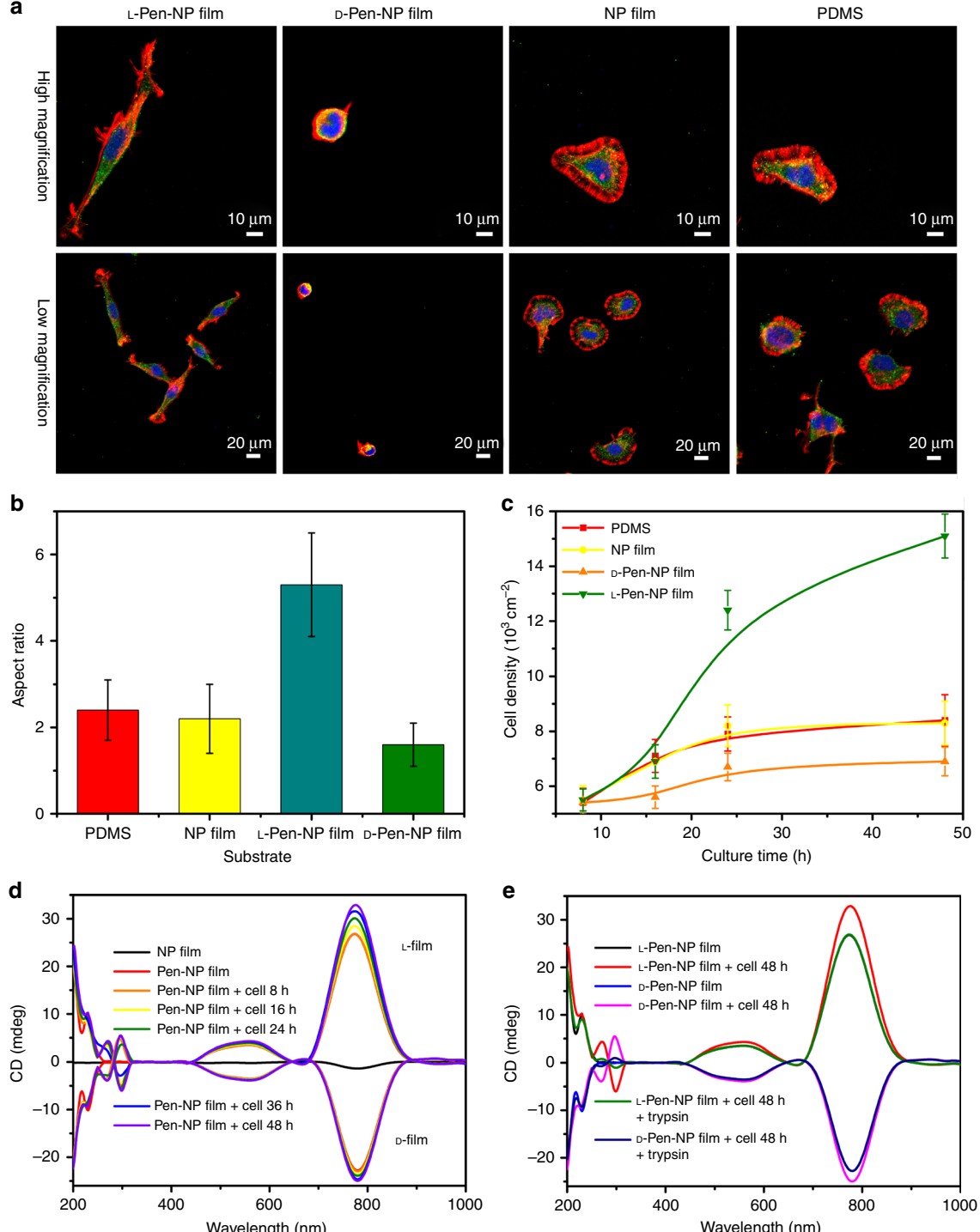

**Fig. 3** Characteristics of NG108-15 cells grown on different substrates. **a** Confocal images of NG108-15 cell grown on different substrates (red, actin; green, vinculin; blue, nucleus) for 8 h. The scale bars of high-magnification images are 10 µm, and scale bars of low-magnification images are 20 µm. **b** Aspect ratio of NG 108-15 cells cultured on different substrates. **c** Cell densities of NG108-15 cells cultured on varies substrates for different time. **d** CD spectra of L- or D-Pen-NP film incubated with NG 108-15 cell for different time. **e** CD spectra of L- or D-Pen-NP film incubated with NG 108-15 cell for 48 h before and after treating with trypsin–EDTA solution. The error bars correspond to the standard error of the mean ($n = 6$)

the distinct cell morphologies on the L- and D-Pen-NP films with different aspect ratio (shape) shown in Fig. 3b, which contributed to vastly different neurite outgrowth after the addition of RA[74, 75]. Third, the twisted spatial conformation of the NPs at the nanoscale provided significant guidance for neurite outgrowth[52, 54], developing the characteristic functions of the cells. Besides the

morphological differentiation, the expression of N-myc onco-protein was proved to be deceased in NG108-15 cells. As shown in Fig. 4b, cells differentiated on L-Pen-NP film exhibited lowest N-myc protein expression, which was consistent with the neurite outgrowth results. Besides the chiral penicillamine, we also applied other chiral molecules such as L/D-cystine, L/D-

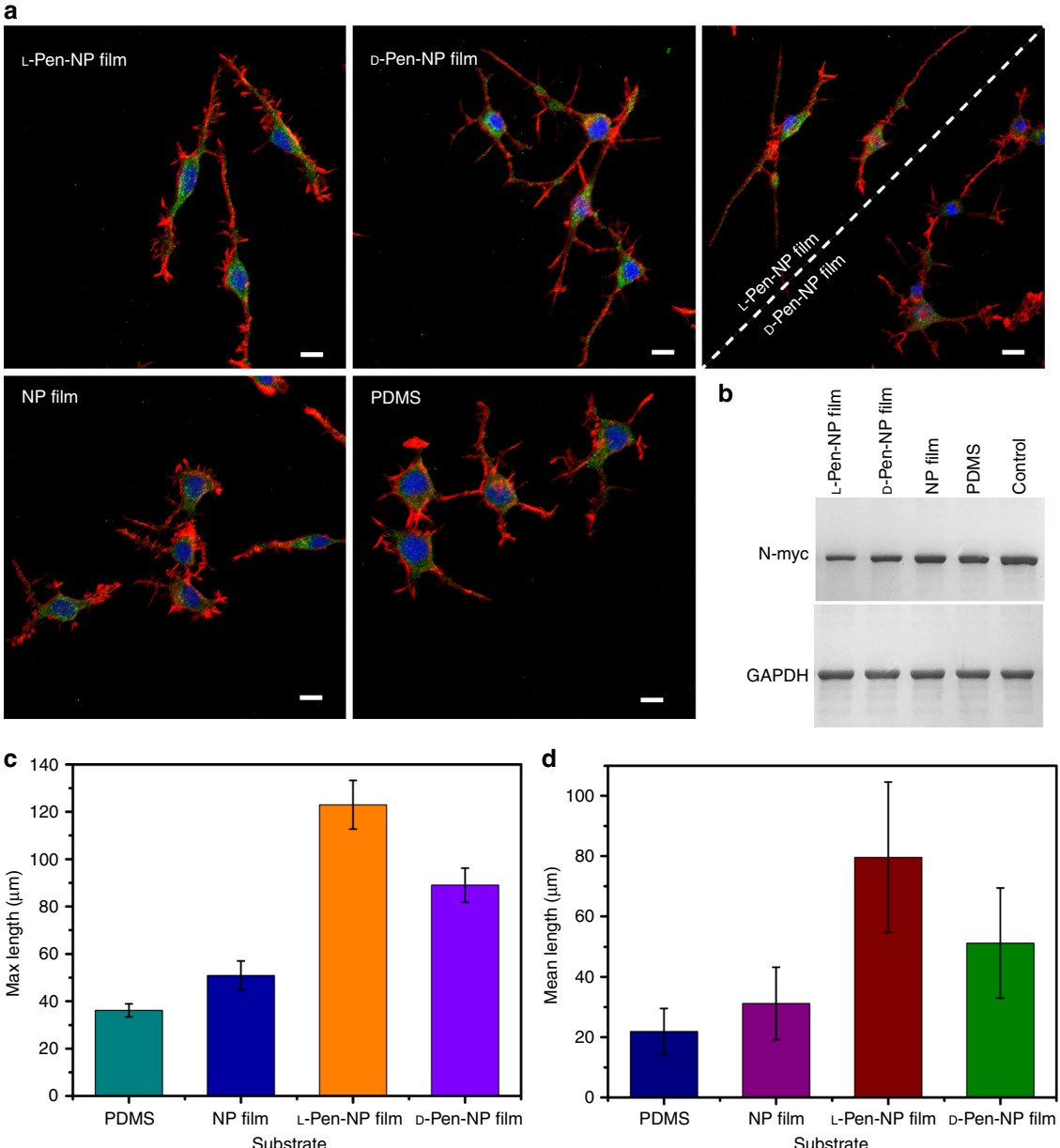

**Fig. 4** Differentiation of NG108-15 cells on different substrates. **a** Confocal images of NG108-15 cells with addition of 1 μM retinoic acid and cultured for 6 days. Scale bar, 20 μm. **b** Expression of N-myc protein in differentiated NG108-15 cells, and GAPDH was used as a reference protein. **c** Max lengths of neurites. **d** Mean lengths of neurites. The error bars correspond to the standard error of the mean ($n = 6$)

phenylalanine to the preparation of chiral NP films, and similar results were obtained (Supplementary Fig. 30). The experimental data of distinct cells differentiation on the chiral films confirmed that the cells interacted with the chiral plasmonic NP film and responded to mechanical stimuli by converting them into biochemical signals, which were probably transmitted through integrin[76]. And the developed L-Pen-NP film might have great potential utility in the repair of sensory damage as bipolar cells could be specialized for the transmission of special senses, such as smell, sight, and hearing.

**Circularly polarized light triggered cell detachment**. Because of the strong plasmon absorption of the Au NP film in the 700–900 nm range, a laser at 808 nm was chosen for the NIR-triggered detachment of the living cells, and NG108-15 cells were selected

as the model. An appropriate safe power level was chosen for the experiment, based on thermal images of the irradiation of different substrates with an 808 nm laser (Supplementary Fig. 31). With the same power density and exposure time (150 mW/cm$^2$, 5 min), the cells detachment on the L-Pen-NP film exposed to left circularly polarized (LCP) light was $91.2 \pm 7.8\%$ and the cells on the D-Pen-NP film exposed to right circularly polarized light displayed $80.7 \pm 7.2\%$ detachment (Fig. 5a, b). In contrast, the control experiment on PDMS resulted in detachment efficiency below 8% (Supplementary Figs. 32 and 33, Fig. 5b). The reason why LCP light led to the high detachment efficiency for cells on L-Pen-NP film was that LCP light activated a larger number of L-Pen-NPs, which increased the number of photoejected hot electrons of chiral film by adsorbing the related rotation of polarized light[67]. Due to a highly efficient cell growth onto the L-Pen-NP film (Figs. 3, 5, and 6, and Supplementary Fig. 17), the L-Pen-NP

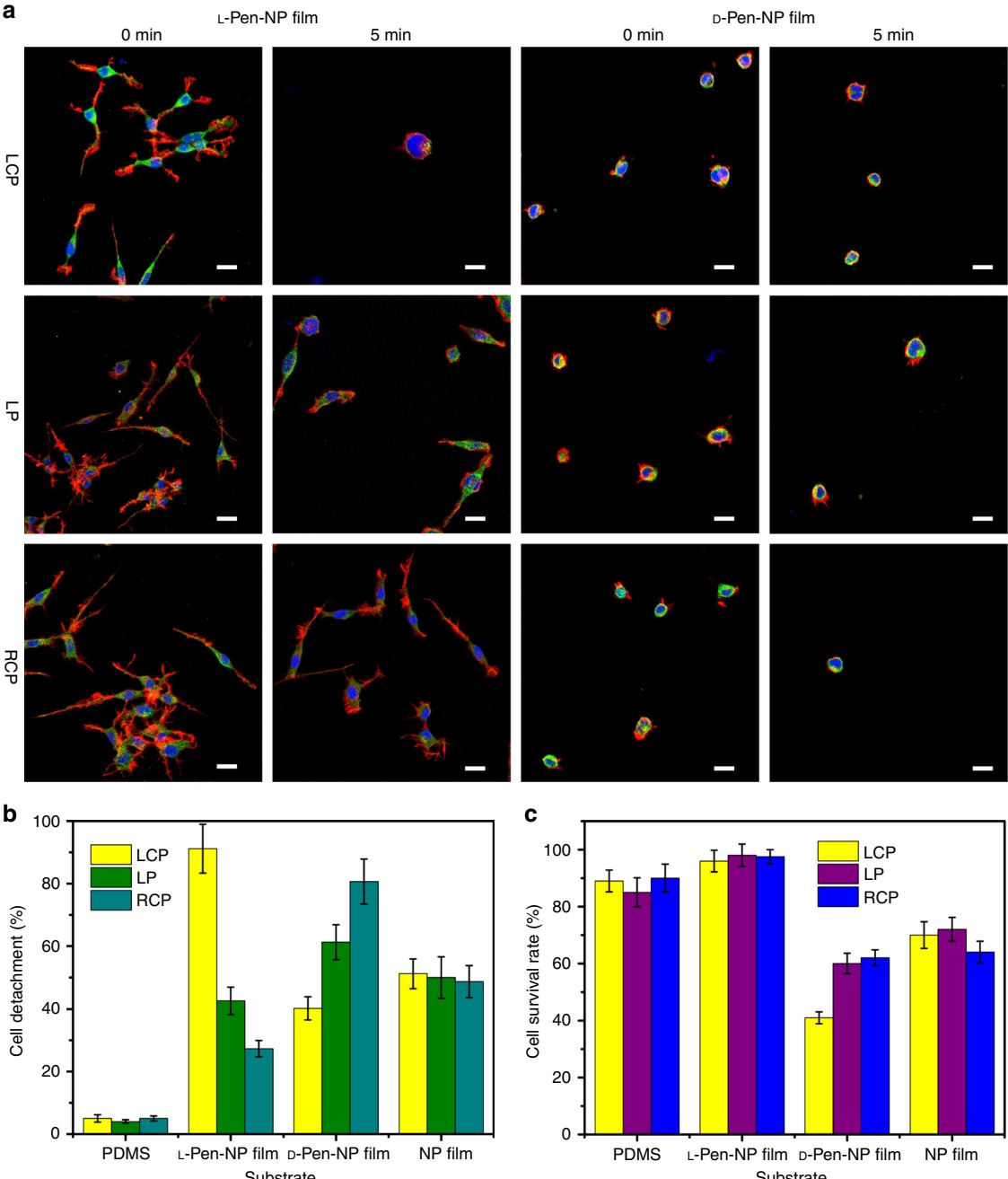

**Fig. 5** Circularly polarized light triggered NG108-15 cells detachment. **a** Confocal images of NG108-15 cell grown on L/D-Pen-NP film upon polarized light irradiation (808 nm laser, 150 mW/cm$^2$). Scale bar, 20 μm. **b** Cell detachment rates upon laser irradiation. **c** Cell survival rates of NG108-15 cells measured by live/dead assay. The error bars correspond to the standard error of the mean ($n = 3$). LCP left circularly polarized light, LP linearly polarized light, RCP right circularly polarized light

film was used as the chiral matrix model to evaluate the relationship between the cell density and detachment efficiencies. This data set exhibited that the cell detachment efficiency was almost independent with the cell density on L-Pen-NP film (Supplementary Figs. 34 and 35). It is well known that NIR laser irradiation can cause cell membrane damage[64]. Therefore, a live/dead assay was conducted on the irradiated cells collected from the substrates. The cells on the L-Pen-NP film retained a living rate exceeding 95%, whereas the viability of the cells on the D-Pen-NP film decreased to 40–60% (Fig. 5c, Supplementary Figs. 36 and 37). Although the cells on PDMS also remained in a good

state, the detachment effect was quite weak (Supplementary Figs. 38 and 39, Fig. 5c). A flow-cytometric analysis (Supplementary Figs. 40–43) was used to confirm the low apoptosis rate (6–8%) of the cells on the L-Pen-NP film. The experiments on the protein adhesion to L- and D-Pen-NP films were also carried out. After 1 h incubation of the films in cell medium (10% fetal bovine serum, FBS), N element was tested via XPS. The result was shown on Supplementary Fig. 44, and the amounts of background N element prior to protein adsorption were very low and similar between L- and D-Pen-NP films. However, after protein adsorption, the amount of N element on L-Pen-NP film was higher than

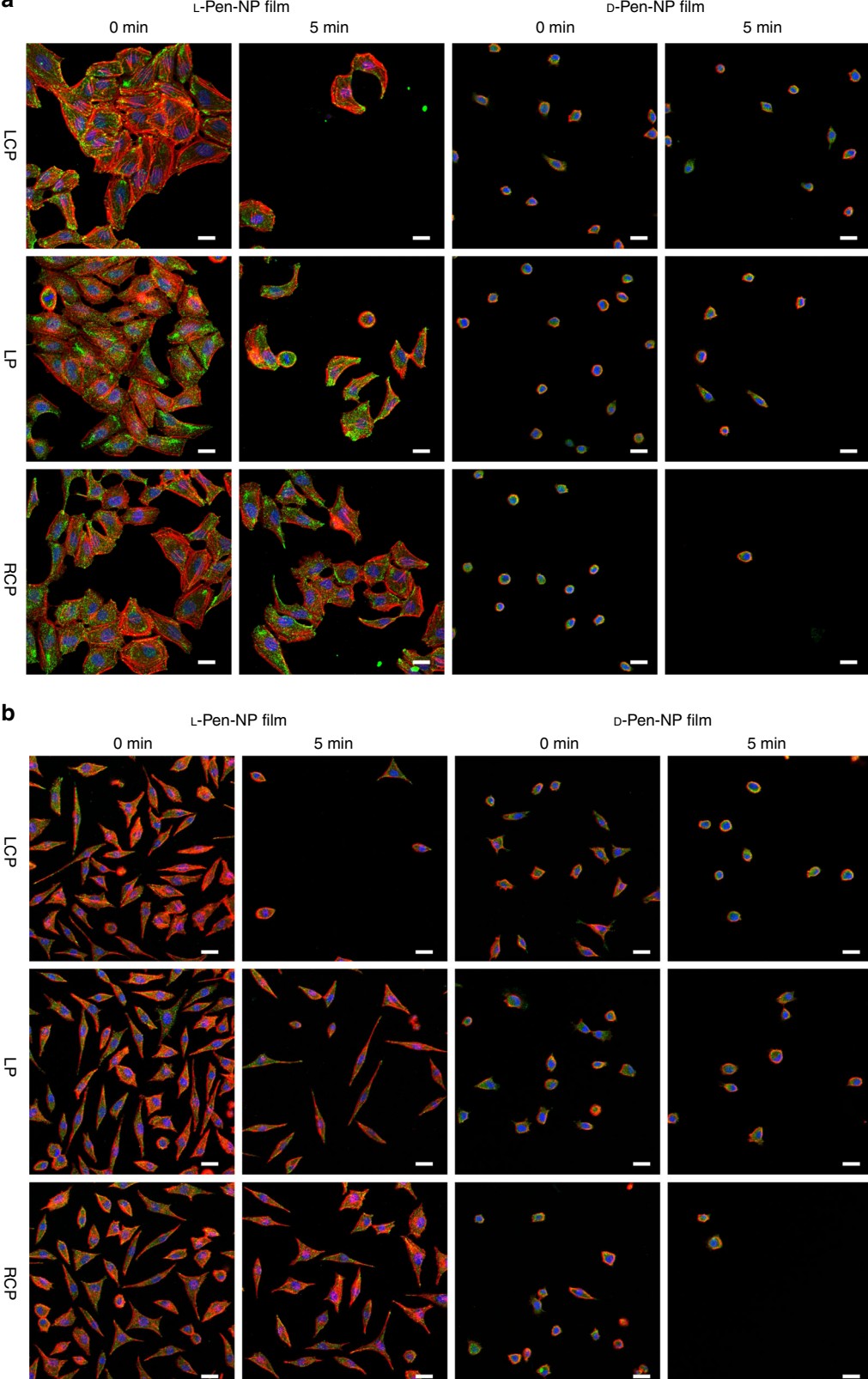

**Fig. 6** Circularly polarized light triggered HeLa and PCS-460-010 cells detachment. Confocal images of (**a**) HeLa cell and (**b**) PCS-460-010 cell grown on L/D-Pen-NP film upon circular polarized light irradiation (808 nm laser, 150 mW/cm$^2$). Scale bar, 25 μm

that on D-Pen-NP film. This point elucidated that more proteins adsorbed on the L-Pen-NP film might work as a protecting layer. Besides, cells on L-Pen-NP film displayed a comparatively larger area than those on D-Pen-NP film. The localized photothermal effect led to the highly efficient detachment of cells, simultaneously avoiding significant damages to cells. Furthermore, the detached cell populations were stable (Supplementary Fig. 45) and remained the ability to differentiate (Supplementary Fig. 46). The cell detachment experiment was also performed on HeLa cells (a typical cancer cell line) and PCS-460-010 cells (normal human primary uterine fibroblasts). The plasmonic film could also be applied to these cell lines (Fig. 6, Supplementary Figs. 47–54), confirming that the L-Pen-NP film provides a highly efficient and safe ECM for cell growth and gentle detachment. Moreover, we detached the differentiated NG108-15 cells on L- or D-Pen-NP film using CPL and then transferred them to the normal tissue culture plate (24 h). The experimental data showed that the detached cells still kept the fine differentiated state (Supplementary Figs. 55 and 56). This useful technique could be applied to manipulate cells for therapy and tissue engineering in the culture system.

In conclusion, the developed monolayer plasmonic NP films with intense chirality remarkably affected cell adhesion, proliferation, and directional differentiation. The different cell behaviors on enantiomeric films were confirmed by the different stereospecific interactions between FN and the films. With the addition of an inducer, NG108 cells differentiated into completely dissimilar neuron-like morphologies on the L- and D-Pen-NP films. These insights could open up an avenue to reveal the chiral self-sorting between the chiral molecules modified film surface and cytomembrane proteins. Noninvasive cell retrieval was achieved with NIR CPL, based on the potent light-polarizing ability of the chiral substrate. It can be anticipated that the application of these effects may speed up the cell culture and engineering, including in the therapeutic use of stem cells. In-depth studies may inspire the design of artificial prosthetic devices and use the biomaterials to recapitulate biological or natural phenomena.

## Methods

**Cell culture**. NG108-15 cells were grown in RPMI-1640 medium supplemented with 10% FBS and 1% penicillin–streptomycin. The cells were passaged by pipetting. For the experiments, the cells were seeded on glass surfaces with or without Au NP films, which were then placed in six-well culture plates. The cells were allowed to grow and adhere to the substrates overnight.

**Cell viability**. NG108-15 cells ($1.0 \times 10^5$) were seeded on PDMS or on PDMS covered with Au NP film (NP film, L-Pen-NP film, or D-Pen-NP film), placed in the wells of a six-well plate, and incubated for 48 h. CCK-8 solution (200 μL) was added to each well and incubated for another 2 h. The absorbance of each well was measured at 450 nm with an Epoch spectrophotometer (BioTek). Relative cell viability (%) was calculated as $(A_{test} / A_{control}) \times 100$.

**FN adsorption**. For the FN adsorption test, 1 mL of FN solution (1000 pg/mL) was added to each well plated with substrate ($1 \times 1$ cm$^2$) in a 24-well culture plate, and the plate was incubated at 37 °C in an atmosphere containing 5% CO$_2$. After the FN had adsorbed, the FN left in the supernatant was quantified with an ELISA kit (Wuhan boster biological technology).

**RA-induced cell differentiation**. NG108-15 cells ($1.0 \times 10^5$) were seeded on PDMS or PDMS covered with Au NP film (NP film, L-Pen-NP film, or D-Pen-NP film), placed in the wells of six-well plates, and incubated for 24 h. After the cells were allowed to adhere to the substrates, RA was added to the cell medium to a final concentration of 1 μM.

**Western blot**. NG108-15 cells ($1.0 \times 10^6$) cultured with cell medium containing RA for 6 days were collected, and protein was extracted by RIPA lysis buffer IV. When the SDS-PAGE electrophoresis was completed, the proteins were transferred to PVDF membrane and then develop the blot according to the protocol (western blot kit, Sangon Biotech Co., Ltd.).

**Fluorescent staining**. NG108-15 cells were grown on PDMS (covered with or without Au NP film), washed in DPBS, fixed with 4% paraformaldehyde for 15 min, treated with 0.25% Triton X-100 for 10 min, and rinsed with DPBS. The cells were incubated with 1.2% bovine serum albumin for 2 h at 37 °C. To stain for vinculin, the cells were incubated with 2 μg/mL anti-vinculin antibody in DPBS with 1% bovine serum albumin for 3 h at room temperature. The samples were then washed three times with ice-cold DPBS, and a goat anti-rabbit IgG (H + L) secondary antibody (2 μg/mL) was added and incubated for 30 min at room temperature. The samples were then washed with ice-cold DPBS. To stain for actin, ActinRed™ 555 ReadyProbes® Reagent (diluted 10-fold in DPBS) was added. After incubation for 30 min at room temperature, the samples were washed again with ice-cold DPBS, stained with DAPI (1/100 in DPBS), and thoroughly washed. Confocal imaging was performed with a Leica TCS SP8 confocal fluorescence microscope. DAPI, green fluorescent protein, and red fluorescent protein filters were used to detect the cell nuclei, vinculin, and actin, respectively.

**Live/dead imaging**. NG108-15 cells subjected to laser treatment were stained with the LIVE/DEAD Cell Imaging Kit (diluted 1/2 in DPBS) for 20 min at room temperature. After staining, the cells were washed three times with ice-cold DPBS. Confocal images were taken with a Leica TCS SP8 confocal fluorescence microscope, with excitation at 488 and 552 nm.

**Data availability**. The authors declare that all data supporting the findings of this study are available within the article and its Supplementary Information files, or are available from the authors upon request.

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

## Acknowledgements

This work is financially supported by the National Natural Science Foundation of China (21771090, 31771084, 21631005, 21673104, 21522102, 21471068 and 21371081).

## Author contributions

H.K. and C.X. conceived the idea and designed the experiments. X.Z., M.S., X.W., and L. X. performed the experiments. W.M. performed simulation. All authors discussed the results, commented on the manuscript, and contributed to the writing of the paper.

## Additional information

**Competing interests:** The authors declare no competing financial interests.

