## [Peer Review File · Nature Communications]

Reviewers' comments:

Reviewer #1 (Remarks to the Author):

Zhao et al examined the effect of material surface chirality on cell behaviors. While this very fundamental topic has been reported in some literature (and has been honestly and explicitly indicated in the manuscript), the present work distinguishes itself as an examination of cells on chiral plasmonic nanoparticle films. The chirality was introduced by linking Au nanoparticles with L- or D-penicillamine.

It is very interesting that such a kind of chiral films exhibited much more significant efficacy than merely chiral gold surfaces and chiral PDMS surfaces. It is also very interesting that the authors examined not only cell adhesion and neutral induction of cells, but also tried to trigger cell detachment using a remotely controlled way by illumination of 808 nm laser.

Since I am interested in this work and I think that the manuscript will be of general interest and high impact, I support its publication after minor revision as follows:

(1) Cell density effect underlying the film chirality effect on cell detachment. Considering that L and D surfaces led to different adhesive densities of cells, the varied detachment efficiencies might be related to the cell density. So, I suggest the authors add data about examination of cell detachment with different densities (under the same chirality). But I do not insist on this revision, if the data are not available now and it will take much time for the authors to do this experiment.

(2) Page 1, lines 11-23, "Because of the broad plasmon absorbance band in the red and near-infrared region, remote irradiation with light (808 nm laser) was chosen to noninvasively harvest the cells. Because the chiral film predominantly absorbs left- or right-handed light, the use of circularly polarized light improved the cell detachment efficiency by up to 91.2% and impaired cell damage." The repeated "because" seems not very nice. May I suggest a change like "Taking advantage of the broad plasmon absorbance band in the red and near-infrared region, remote irradiation with light (808 nm laser) was chosen to noninvasively harvest the cells? Because the chiral film predominantly absorbs left- or right-handed light, the use of circularly polarized light improved the efficiency of cell detachment by up to 91.2% and impaired cell damage."

(3) Page 2, lines 4-6, "Most cell functions, such as adhesion,10–12 proliferation,13–14 and differentiation,15–19 are affected by the interactions between the cells and the extracellular matrix (ECM).20–26" Here, "dedifferentiation" could be mentioned (see, for instance, *Biomaterials*, 35, 6871-6881 (2014); *Nano Lett.*, 15(11), 7755-7765 (2015). At the end of this sentence, some other recent pertinent publications (for instance, *Biomaterials*, 111, 27-39 (2016); *ACS Appl. Mater. Interfaces*, 8, 21903–21913 (2016)) could also be mentioned, which might be helpful for the readers to know the background better.

(4) Page 3, line 19, "as a photoswitch to control cell detachment,56 And Liz-Marzan and coworkers" Here, ", " ahead of 56 should be ".".

(5) Page 4, lines 5-7. "Here, we designed monolayer chiral Au NP films for use in cell adhesion, growth, and differentiation. As illustrated in Figure 1A, NG108-15 cells were seeded on L- or D-NP films supported by polydimethylsiloxane (PDMS), at the same initial density." Here, I suggest the authors explain explicitly the meanings of L-NP and D-NP in order to enhance readability.

(6) Page 8, after the first paragraph about Figure 4 about the distinct differentiation states of NG108-15 cells on the chiral films, the authors might add a paragraph to give a brief discussion about some

potential reasons of the different differentiation. The varied cell adhesive shape with the evidence of the aspect ratios shown in Figure 3b might be one of potential reasons. See, for instance, *Biomaterials*, 32, 8048-8057 (2011); *Biomaterials*, 34(4), 930-939 (2013). Although these publications describe osteogenic and adipogenic differentiations of mesenchymal stem cells with a different topic, they unambiguously indicated that cell adhesive shape influences later cell differentiation. So, the shape factor might be one of potential reasons of different differentiation in the present work.

(7) Reference: The "Acs Nano" might be "ACS Nano".

(8) Page 22, caption of Figure 5. I suggest explain "LCP", "LP" and "RCP" in the caption in order to improve readability.

Reviewer #2 (Remarks to the Author):

Authors described the plasmonic chiral Au nanoparticle (NP) film modified with L/D-penicillamine for chirality controlled neuronal cell growth and differentiation.

They also demonstrated the NIR-triggered detachment of the living cells using left circularly polarized light for the cells on the L-NP film and right circularly polarized light for the cells on the D-NP film.

Eventhough this presented work is very well written and described; similar single ideas/studies e.g. use of L/D-penicillamine functionalized nanomaterial monolayer, chirality dependent Neuronal Growth and stem cell differentiation, effect of protein adsorption were reported individually elsewhere.

For example,

-Cell interactions with a L/D-penicillamine functionalized nanomaterial monolayer has been reported previously by Kehr et al. *Angew. Chem. Int. Ed.*, 2012, 51, 3716-3720

-Neuronal Growth on L- and D-Cysteine Self-Assembled Monolayers was described by Shefi et al. *ACS Chem. Neurosci.* 2014, 5, 370–376

-Effects of surface molecular chirality on adhesion and differentiation of stem cells was reported by Ding et al. *Biomaterials* 2013, 34, 9001–9009.

-Additionally, effect of protein adsorption on chiral surfaces and NP surfaces and its subsequent effects on cell behavior are known from the previous reports as well.

Adv. Funct. Mater. 2011, 21, 3276–3281

Colloids and Surfaces B: Biointerfaces 2012, 90, 97–101

Biomacromolecules, 2016, 17, 1117.

RSC Adv. 2015, 5, 5704.

Mater. Chem. Front. 2017, DOI:10.1039/c6qm00163g.

Sci. Rep. 2016, 6, 31595.

Macromol. Biosci. 2014, 14, 793.

Therefore for my opinion this work is not novel enough for Nature Communication. However the idea "the irradiation of the chiral films at near-infrared region and the subsequent absorption of left- or

right-handed light and the use of circularly polarized light for improved cell detachment" is new and interesting part of this study. Unfortunately, this part is not very well discussed in the manuscript.

My other comments:

-The discussion on the plasmonic chirogenesis in page 5 line 109-115 is not clear. Authors should explain this part better.

- Authors used fibronectin (I assume as a representative example of protein) to investigate the stereospecific interaction between cells and enantiomeric NP films. Then they claimed that (page 7 line 138-140) "Therefore, the distinct cell morphologies and proliferation behaviors of cells on enantiomeric NP films might be attributable to the different signals released from the stereospecific interactions between FN and the chiral NP films (Figure 1B)."

Why did they used fibronectin? And how do they know that fibronectin is the result of this observed effect because cell experiments were done in serum (FBS) containing media and in FBS there are many other proteins available as well.

-Authors mentioned that (page 7 line 145-152) "The CD signals of the chiral NP film were also astonishingly enhanced after cell adhesion and proliferation ... of the film in the Vis-NIR region". But they did not discuss what most likely reason is?

For example they can discuss that cells has intrinsic chirality maybe this is the reason of the enhanced CD signals etc.

-Authors mentioned that (page 9, line 189-195) "The cells on the L-NP film retained a living rate exceeding 95%, whereas the viability of the cells on the D-NP film decreased to 40%–60% (Figure 5C, S28–29). Although the cells on PDMS also remained in a good state, the detachment effect was quite weak (Figure S30–33). A flow-cytometric analysis (Figure S34–37) was used to confirm the low apoptosis rate (6%–8%) of the cells on the L-NP film."

Again Authors should discuss more in detail the most likely reason of their observation (The result indicated that the left-handed film might have a protective effect on cells against the NIR laser as special ECM).

For example since more proteins adsorbed on the surface of L-enantiomer this may work as a protecting layer etc.

-I think the reason of the use of NIR circularly polarized light (CPL) to improve the efficacy of light in cell detachment was not very well described eventhough this is the most innovative and interesting part of this study.

Reviewer #3 (Remarks to the Author):

– This paper introduces a nice platform innovation for cell-material interplay leading to increased proliferation, directed differentiation and for cell retrieval using the unique properties of surface chiral configuration and architectures. The study is significant in that it addresses an important and biologically relevant topic that has been overlooked in cell manipulation techniques and technology as well as in the design of biomaterials. For me the study requires more breadth and depth to be accepted for publication in Nature Communications.

→ I do not find the application of the chiral films convincing enough to warrant publication in Nature Communications as they stand. The study is technically sound and accurate, but is a little too limited for the Nature Communications forum. There are large limitations in the cell studies. Principally it needs to go further in the process and analysis. The materials innovation is good and the materials work is high quality and tight.

→ This is an interesting study in parts, but it remains incohesive; meaning that I am unclear about its primary focus and the significance it has for the stated aims, which were stated as tissue engineering and cell manipulation. More explanation is needed on the problem that can be tackled with this technology. I would like to see more evidence of the cell growth and differentiation into structures with biotechnological value. I think the neurite morphology shape change on the chiral surfaces is really interesting. Cells that are grown on this platform are preferentially differentiated on the L-form surface and detached. Are the detached cell populations stable and do they retain their differentiated state?

→ The tuning effect with the plasmon film is novel and interesting too, but much more detail on its effect on cells, collective cell behaviour and applications are needed.

→ The abstract. The first sentence is correct but it is also important in biology and medicine too.

→ I am confused at the start on the specific application of the chiral film? You should state clearly that the technology is designed to promote a cell response in type and numbers perhaps, and be able to easily remove them for other use. I am not clear on the removal mechanism that you have invented.

→ I liked the idea that the chiral surface facilitated the detachment of cells using the light irradiation method. That is useful technique to have when manipulating cells for therapy and tissue engineering in the culture laboratory. Maybe you could emphasize this. Use your platform for driving cells into a fate and into morphologies and populations that form volumes or functional structures that can be transferred to another culture system or be implanted?

→ What is the ultimate destination for this technology in biomedicine? Please add specific applications in this regard.

→ I don't like the last sentence as it is too general. Many other studies have demonstrated the utility of biomaterials to recapitulate biological/ natural phenomena. Also I did not get the sense that this study "clarified chiral selection" more it added to the body of evidence showing that chirality influences cell behaviour, in this case neurite growth. Infact the cell behaviours were altered by the combination of chiral patterned surface and chemical inductor.

→ The Introduction needs improving. It needs to have more rigid cohesion and more breadth. I would particularly be interested to read a stronger rationale for designing artificial ECM with chirality. I am with you on its significance as all biological systems have chiral

→ Maybe you could identify the chiral features of ECM that have been reported to support your statements on the importance of ECM-cell interactions. I would like to read about the studies investigating this complementarity.

→ Please include, albeit in other ways, that materials have been designed and fabricated to add chiral features and signatures. There is an important set of literature on chiral supramolecular biomaterials.

- Detail to add on bland statement: "These studies have shown that cells can interact with chiral surface molecules and display different behaviors on enantiomorphous surfaces"
- Make a case as to why nanoparticles attached to a 2D platform would be used as an artificial ECM system?
- It would be nice to know whether the neurites from synaptic connections and can be proved to transmit electrical signals while attached onto this special surface.
- I would suggest a clearer description of the aims of your study. The introduction is a confusing preface to this study. I was led to think at the start that the aim was to fabricate ECM mimics for cell biocompatibility. Then we moved towards a technical device for cell retrieval.
- The results are interesting. So, we now know that your chiral surfaces promote neurite cells in favourable ways.
- Use of headings in the script will be helpful.
- You state that the study could lead onto better understanding of chiral selectivity, so can you adapt your fabrication to create lots of chiral variants? Can you replicate the chiral signatures seen in nature onto your platform? How versatile is the platform to do this and work for biotechnology, cell therapy requirements? I think you have a nice platform technology but more work is needed to show its good potential. For me the studies layed down do not do that.
- It would be important to extend the cell studies beyond 5 minutes to realise the potential of the surfaces on proliferation, differentiation responses. The timing should be extended to determine that the cells are stable, stably attached and whether they process the new environment and develop into overt tissue structures, for instance.
- You could have used more display items of up to 10 to increase the depth and breadth of the study material.
- I would like to see more highly resolved observation of the cell interaction mechanics with the chiral architecture and Fn associations and its conformation. You state that the film is an ECM.
- More evidence is needed to justify the claim that the nanosurfaces can be used to "clarify the chirality selection of biosystems".

Reviewer #4 (Remarks to the Author):

The application of chiral surfaces is quite important in biomedical treatment and clinical detection. This manuscript prepared chiral penicillamine-modified nanoparticle coating PDMS films to control cell adhesion, differentiation and detachment. The results are very interesting. However, some problems in the writing, experiment and conclusion make it not well understood. Resubmission after more careful researches is suggested.

1. There are a series of materials. However, they were named not systematically and given the definition from the beginning. For example, L-NP film and D-NP film in the manuscript had been also named as L-PEN NP film and D-PEN NP film. While the methods for L-/D-Au coating and L-/D-PDMS

were not given. By the way, it is not clear if L-/D-PEN were synthesized by the authors or from any commercial product.

2. The bond formed between PEN and AuNP should be Au-S. However, it looks like just physical adsorption from Figure 2B.

3. The Au nanoparticles were observed at the size of around 50 nm. As expected, after modified by PEN, their sizes would be a little bigger. However, no direct evidence was provided. One can read from Figure 2C and 2D with a conclusion about 30-40 nm, which is different from the results of Figure S1 and the sentence in Line 22 on Page 4.

4. The property of the films should be not just due to their chirality, but also the modified amount of L-/D-PEN. However, there is no result provided. To evaluate their effects on cell behaviors, they should be controlled at the same levels on the films. More importantly, different levels of L-/D-PEN on the surfaces can give the significant effects and trends on cell adhesion and differentiation.

5. Moreover, although FN adsorption increased on L-PEN NP film, it couldn't give the explanation of the cellular morphology and neural differentiation. The mechanism should be investigated more carefully.

6. The illumination at 808 nm will promote cell detachment. While the mechanism is due to high temperature treatment. From Figure S25, the temperature of L-NP film could reach to almost 60 degree after 5 min. It's really an extreme condition for cell survival. And why the other materials and conditions just reached a lower temperature after laser excitation but with a higher cell death? How can L-PEN protect the cells?

Response to Reviewers

Reviewers' comments:

Reviewer #1 (Remarks to the Author):

Comments 1-1: Zhao et al examined the effect of material surface chirality on cell behaviors. While this very fundamental topic has been reported in some literature (and has been honestly and explicitly indicated in the manuscript), the present work distinguishes itself as an examination of cells on chiral plasmonic nanoparticle films. The chirality was introduced by linking Au nanoparticles with L- or D-penicillamine.

It is very interesting that such a kind of chiral films exhibited much more significant efficacy than mealy chiral gold surfaces and chiral PDMS surfaces. It is also very interesting that the authors examined not only cell adhesion and neutral induction of cells, but also tried to trigger cell detachment using a remotely controlled way by illumination of 808 nm laser.

Since I am interested in this work and I think that the manuscript will be of general interest and high impact, I support its publication after minor revision as follows:

Reply: We appreciate your hard work to review our manuscript and give the positive feedback, and we provide in the following a detailed point by-point response to the raised questions.

Comments 1-2: Cell density effect underlying the film chirality effect on cell detachment. Considering that L and D surfaces led to different adhesive densities of cells, the varied detachment efficiencies might be related to the cell density. So, I suggest the authors add data about examination of cell detachment with different densities (under the same chirality). But I do not insist on this revision, if the data are not available now and it will take much time for the authors to do this experiment.

Reply: We completely agree with this point.

As suggested by Reviewer 1, we adjusted the incubation time of chiral substrates and cells to obtain the diverse cell density, and then, compared the cell detachment efficiencies under the same chirality. As a highly efficient for cell growth onto the L-Pen-NP film (**Figure 3, 5, 6, S17**), we chose the L-Pen-NP film as the chiral matrix model to evaluate the detachment efficiencies.

As increasing the incubation time, the cell density onto the L-Pen-NP film could be gradually increased (**Figure 3C, S17**). And also, we used the CD spectra to real-time monitor the chiroptical activities of NP films. Owing to the intrinsic chirality of cells¹⁻³, we can see the CD signal in the 700-900 nm range could be slightly enhanced as increasing the incubation time, which suggested the amount of cells onto the L-Pen-NP film was progressively elevated (**Figure R1**). This is accordance with the confocal data.

The NG108-15 cells ($(0.54 \pm 0.05) \times 10^4/\text{cm}^2$) were seeded on L-Pen-NP film.

We further calculated the cell densities of NG108-15 cultured on L-Pen-NP film for 16 h and 24 h were $(0.69 \pm 0.06) \times 10^4/\text{cm}^2$ and $(1.24 \pm 0.07) \times 10^4/\text{cm}^2$, respectively, which were lower than that of 48 h ($(1.51 \pm 0.08) \times 10^4/\text{cm}^2$) (**Figure R2, Figure 5A**). At the various incubation times, we illuminated the NG108-15 cells with L-Pen-NP film by 880 nm light under different polarization conditions (150 mW/cm², 5 min). These photons match the strongest CD band of the chiral NP films. And, the chiroplasmonic band is positive for the complex of NP film and adhesive cells (**Figure 3D**). And we also found that the detachment efficiencies were almost same under the various incubation time (**Figure R3**). The results were 90.5 ± 7.2 % under LCP, 43.3 ± 5.2 % under LP, and 27.4 ± 2.8 % under RCP, respectively (Mean values). It was shown that left-handed circularly polarized light (LCP) gives a three-fold higher in cell detachment efficiencies compared with right-handed circularly polarized light (RCP) (**Figure 5, R2 and R3**). This suggested that the cell detachment efficiency was almost independent with the cell density on L-Pen-NP film.

Moreover, With the same power density and exposure time (150 mW/cm², 5 min), the cells detachment on the L-Pen-NP film (48 h) exposed to left circularly polarized (LCP) light was 91.2 ± 7.8 % and the cells on the D-Pen-NP film (48 h) exposed to right circularly polarized (RCP) light displayed $80.7 \pm 7.2\%$ detachment (**Figure 5A, 5B**).

The reason why LCP light led to the high detachment efficiency for cells on L-Pen-NP film was that LCP light activated a larger number of L-Pen-NPs than other light conditions, which increased the number of photoejected hot electrons of chiral film by adsorbing the related rotation of polarized light.

The reason why LCP light led to the high detachment efficiency for cells on L-Pen-NP film was that LCP light activated a larger number of L-Pen-NPs than other light conditions, which increased the number of photoejected hot electrons of chiral film by adsorbing the related rotation of polarized light.

Due to a highly efficient for cell growth onto the L-Pen-NP film (Figure 3, 5, 6, S17), the L-Pen-NP film was used as the chiral matrix model to evaluate the detachment efficiencies (Figure S34-35). The data exhibited that the cell detachment efficiency was almost independent with the cell density on L-Pen-NP film.

Figure R1. CD spectra of L-Pen-NP film incubated with NG 108-15 cell for different time.

Figure R2 (Figure S34). Confocal images of NG108-15 cells grown on L-Pen-NP film for 16, 24 and 48 h, upon polarized light irradiation (808nm laser, 150 mW/cm²). (Red, actin; green, vinculin; blue, nucleus). Scale bars were 20 μm.

Figure R3 (Figure S35). Cell detachment rates after laser irradiation. NG108-15 cells, after cultured on L-Pen-NP film for 48 (as in **Figure 5A**), 24 and 16 h (as in **Figure R2**), and then were detached under the illumination of near-infrared light (808nm laser, 150 mW/cm², 5 min).

Comments 1-3: Page 1, lines 11-23, “Because of the broad plasmon absorbance band in the red and near-infrared region, remote irradiation with light (808 nm laser) was chosen to noninvasively harvest the cells. Because the chiral film predominantly absorbs left- or right-handed light, the use of circularly polarized light improved the

cell detachment efficiency by up to 91.2% and impaired cell damage.” The repeated “because” seems not very nice. May I suggest a change like “Taking advantage of the broad plasmon absorbance band in the red and near-infrared region, remote irradiation with light (808 nm laser) was chosen to noninvasively harvest the cells? Because the chiral film predominantly absorbs left- or right-handed light, the use of circularly polarized light improved the efficiency of cell detachment by up to 91.2% and impaired cell damage.”

Reply: Thanks a lot. We have corrected the sentence in the revised version as your nice suggestion.

Comments 1-4: Page 2, lines 4-6, “Most cell functions, such as adhesion,^{10–12} proliferation,^{13–14} and differentiation,^{15–19} are affected by the interactions between the cells and the extracellular matrix (ECM).^{20–26}” Here, “dedifferentiation” could be mentioned (see, for instance, *Biomaterials*, 35, 6871–6881 (2014); *Nano Lett.*, 15(11), 7755–7765 (2015). At the end of this sentence, some other recent pertinent publications (for instance, *Biomaterials*, 111, 27–39 (2016); *ACS Appl. Mater. Interfaces*, 8, 21903–21913 (2016)) could also be mentioned, which might be helpful for the readers to know the background better.

Reply: many thanks for your nice comments. We have added the related information and references into the revised version as follows:

Most cell functions, such as adhesion,^{10–12} proliferation,^{13,14} and differentiation,^{15–19} dedifferentiation,^{20,21} are affected by the interactions between the cells and the extracellular matrix (ECM).^{22–30}

20. Li, S. *et al.* Effects of nanoscale spatial arrangement of arginine-glycine-aspartate peptides on dedifferentiation of chondrocytes. *Nano Lett.* **15**, 7755–7765 (2015).
21. Cao, B., Peng, R., Li, Z. & Ding, J. Effects of spreading areas and aspect ratios of single cells on dedifferentiation of chondrocytes. *Biomaterials* **35**, 6871–6881 (2014).
29. Ye, K., Cao, L., Li, S., Yu, L. & Ding, J. Interplay of Matrix Stiffness and Cell-Cell Contact in Regulating Differentiation of Stem Cells. *ACS Appl. Mater. Interfaces* **8**, 21903–21913 (2016).
30. Liu, X. *et al.* Subcellular cell geometry on micropillars regulates stem cell differentiation. *Biomaterials* **111**, 27–39 (2016).

Comments 1-5: Page 3, line 19, “as a photoswitch to control cell detachment,⁵⁶ And Liz-Marzan and coworkers” Here , ”,” ahead of 56 should be “.”.

Reply: Thanks a lot. It was corrected.

Comments 1-6: Page 4, lines 5-7. “Here, we designed monolayer chiral Au NP films for use in cell adhesion, growth, and differentiation. As illustrated in Figure 1A, NG108-15 cells were seeded on L- or D-Pen-NP films supported by polydimethylsiloxane (PDMS), at the same initial density.” Here, I suggest the authors explain explicitly the meanings of L-Pen-NP and D-Pen-NP in order to enhance readability.

Reply: Thanks a lot. We explained the explicit meanings of L-Pen-NP and D-Pen-NP in the revised version as follows.

As illustrated in Figure 1A, NG108-15 cells were seeded on L- or D-Pen-NP (Au nanoparticles modified with L- or D-penicillamine) films supported by polydimethylsiloxane (PDMS), at the same initial density.

Comments 1-7: Page 8, after the first paragraph about Figure 4 about the distinct differentiation states of NG108-15 cells on the chiral films, the authors might add a paragraph to give a brief discussion about some potential reasons of the different differentiation. The varied cell adhesive shape with the evidence of the aspect ratios shown in Figure 3b might be one of potential reasons. See, for instance, *Biomaterials*, 32, 8048-8057 (2011); *Biomaterials*, 34(4), 930-939 (2013). Although these publications describe osteogenic and adipogenic differentiations of mesenchymal stem cells with a different topic, they unambiguously indicated that cell adhesive shape influences later cell differentiation. So, the shape factor might be one of potential reasons of different differentiation in the present work.

Reply: Thank you very much for your very useful suggestions. As one of the potential reasons of different differentiation behaviors of cells on chiral substrates, the shape factor⁴⁻⁵ has been added and discussed in revised version.

Second, the stereospecific interaction between FN and the chiral NP films caused the distinct cell morphologies on the L- and D-Pen-NP films with different aspect ratio (shape) shown in Figure 3B, which contributed to vastly different neurite outgrowth after the addition of RA.⁷⁴⁻⁷⁵

(74)Peng, R.; Yao, X.; Ding, J., Effect of cell anisotropy on differentiation of stem cells on micropatterned surfaces through the controlled single cell adhesion,

Biomaterials **2011**, *32*, 8048.

(75) Yao, X.; Peng, R.; Ding, J., Effects of aspect ratios of stem cells on lineage commitments with and without induction media, *Biomaterials* **2013**, *34*, 930.

Comments 1-8: Reference: The “Acs Nano” might be “ACS Nano”.

Reply: Thanks a lot. It was corrected. And we have thoroughly checked the style of the remaining references to satisfy the requirement by Nature Communications.

Comments 1-9: Page 22, caption of Figure 5. I suggest explain “LCP”, “LP” and “RCP” in the caption in order to improve readability.

Reply: Many thanks. In this paper, “LCP”, “LP” and “RCP” were referred as “left-handed circular polarized light”, “linearly polarized light”, and “right-handed circular polarized light”, respectively. And we have added the detailed explanation about “LCP”, “LP” and “RCP” in the caption of Figure 5 in the revised version.

Revision in the revised version:

Figure 5. **A)** Confocal images of NG108-15 cell grown on L/D- Pen-NP film upon polarized light irradiation (808nm laser, 150 mW/cm². LCP: left-handed circularly polarized light; LP: linearly polarized light; RCP: right-handed circularly polarized light). Scale bar, 20 μm. **B)** Cell detachment rates upon laser irradiation. **C)** Cell survival rates of NG108-15 cells measured by Live/dead assay.

Reviewer #2 (Remarks to the Author):

Comments 2-1: Authors described the plasmonic chiral Au nanoparticle (NP) film modified with L/D-penicillamine for chirality controlled neuronal cell growth and differentiation.

They also demonstrated the NIR-triggered detachment of the living cells using left circularly polarized light for the cells on the L-Pen-NP film and right circularly polarized light for the cells on the D- Pen-NP film.

Eventhough this presented work is very well written and described; similar single ideas/studies e.g. use of L/D-penicillamine functionalized nanomaterial monolayer,

chirality dependent Neuronal Growth and stem cell differentiation, effect of protein adsorption were reported individually elsewhere.

For example,

-Cell interactions with a L/D-penicillamine functionalized nanomaterial monolayer has been reported previously by Kehr et al. *Angew. Chem. Int. Ed.*, 2012, 51, 3716-3720

-Neuronal Growth on L- and D-Cysteine Self-Assembled Monolayers was described by Shefi et al. *ACS Chem. Neurosci.* 2014, 5, 370–376

-Effects of surface molecular chirality on adhesion and differentiation of stem cells was reported by Ding et al. *Biomaterials* 2013, 34, 9001–9009.

-Additionally, effect of protein adsorption on chiral surfaces and NP surfaces and its subsequent effects on cell behavior are known from the previous reports as well.

Adv. Funct. Mater. 2011, 21, 3276–3281

Colloids and Surfaces B: Biointerfaces 2012, 90, 97–101

Biomacromolecules, 2016, 17, 1117.

RSC Adv. 2015, 5, 5704.

Mater. Chem. Front. 2017, DOI:10.1039/c6qm00163g.

Sci. Rep. 2016, 6, 31595.

Macromol. Biosci. 2014, 14, 793.

Reply: Thank you very much for your hard work to our manuscript.

We summarized the above literatures here:

1. Cell Adhesion Behavior on Enantiomerically Functionalized Zeolite L Monolayers. *Angew. Chem. Int. Ed.*, 2012, 51, 3716-3720: L/D PEN-functionalized zeolite monolayers were designed for cell adhesion, demonstrating that cells could recognize and differentiate between the different enantiomorphous zeolite surfaces. Cells adhered on L surface were higher than those on D surface.
2. Neuronal Growth on L- and D-Cysteine Self-Assembled Mono layers Reveals Neuronal Chiral Sensitivity. *ACS Chem. Neurosci.* 2014, 5, 370–376: L/D cysteine monolayers on Au films (deposited on mica substrates using a high vacuum sputtering technique) for neuronal growth. The L-Cys coating reduces the

ability of neurons to attach to the surface and to develop optimized branching trees. These results are opposed to previous studies of the interaction of non-neuronal cells with Cys-treated surfaces.

3. Effects of surface molecular chirality on adhesion and differentiation of stem cells. *Biomaterials* 2013, 34, 9001–9009.: Molecular chirality influenced stem cells. Higher density of cell adhesion and larger cell spreading area on L-Cys substrate, with higher cell tension in turn favored osteogenesis rather than adipogenesis. Additionally, effect of protein adsorption on chiral surfaces and NP surfaces and its subsequent effects on cell behavior are known from the previous reports as well.
4. Chiral Design for Polymeric Biointerface: The Influence of Surface Chirality on Protein Adsorption. *Adv. Funct. Mater.* 2011, 21, 3276–3281: Chiral self-assembly monolayers of L/D-valine on gold surface. L-film has much stronger affinity with proteins than the corresponding D-film.
5. Cell adhesion on chiral surface: The role of protein adsorption. *Colloids and Surfaces B: Biointerfaces* 2012, 90, 97–101: Gold surfaces modified with L-Cys supported more cell adhesion than those modified with D-Cys. More serum proteins adsorbed onto the L-Cys modified surfaces than the D-Cys modified one.
6. Enantiomorphous Periodic Mesoporous Organosilica-Based Nanocomposite Hydrogel Scaffolds for Cell Adhesion and Cell Enrichment. *Biomacromolecules*, 2016, 17, 1117: Cells recognized and differentiated differently between enantiomers of chiral amino acid or carbohydrate functionalized nanocomposite hydrogel scaffolds. It was used to enrich one cell type from a heterogeneous mixture of two cells.
7. Self-assembled monolayers of enantiomerically functionalized periodic mesoporous organosilicas and the effect of surface chirality on cell adhesion behavior. *RSC Adv.* 2015, 5, 5704: D- and L-mannose modified monolayer of mesoporous organosilicas were used to monitor the different cell adhesion behavior to separate malignant HeLa or C-6-Glioma cell lines from healthy endothelial cells.

8. Influence of protein adsorption on the cellular uptake of AuNPs conjugated with chiral oligomers. *Mater. Chem. Front.* 2017, DOI:10.1039/c6qm00163g: Influence of protein adsorption on the cellular uptake of AuNPs conjugated with chiral oligomers. The high adsorption of serum proteins on the NP surface could be used to screen the PAV molecules from being recognized by the cells after incubation in fetal bovine serum with a high concentration (50%), and thereby L- and D-PAV–AuNPs did not show a significant difference.
9. Surface-anchored poly (acryloyl-L (D)-valine) with enhanced chirality-selective effect on cellular uptake of gold nanoparticles. *Sci. Rep.* 2016, 6, 31595: cellular uptake being triggered by chiral polymers-capped NPs. poly(acryloyl-D-valine)-AuNPs were internalized with significantly larger amount than the L- AuNPs.
10. Cell Adhesion Behavior in 3D Hydrogel Scaffolds Functionalized with D- or L-Aminoacids. *Macromol. Biosci.* 2014, 14, 793: D- or L-Amino acids functionalized hydrogel scaffolds for cell adhesion. Different cell lines recognize and differentiate the respective hydrogel scaffolds, resulting in a higher affinity of alive C-6-glioma and endothelial cells to D-PEN-Alg scaffold than to L-PEN-Alg scaffold.

The significances of our text are as follows.

1. Monolayer chiral Au nanoparticle (NP) film with circular dichroism (CD) peaks at 550 nm and 775 nm was first fabricated for cell growth, differentiation and retrieval.
2. It was found that L-Pen-NP film could accelerate cell proliferation while D-Pen-NP film exhibited opposite effect, which could be ascribed to stereospecific interaction between fibronectin and chiral NP film.
3. The twisted spatial conformation of NP at nanoscale provided significant guidance in neurite outgrowth. Cells on L-Pen-NP film displaying dipolar differentiation, whereas multipolar neurite outgrowth occurred on D- Pen-NP film.
4. Due to the broad plasmon absorbance band in visible and near-infrared region, circularly polarized light (808 nm laser) was conducted for high efficiency non-invasive cell harvest.

Some related references above were cited in the text.

Comments 2-2: Therefore for my opinion this work is not novel enough for Nature Communication. However the idea “the irradiation of the chiral films at near-infrared region and the subsequent absorption of left- or right-handed light and the use of circularly polarized light for improved cell detachment” is new and interesting part of this study. Unfortunately, this part is not very well discussed in the manuscript.

Reply: Thank you for your comments very much.

We are glad to know that you enjoy our experiment design of cell detachment by NIR circularly polarized light. Also, we would like to discuss more details and the potentials of the designed ECM. The controlled and noninvasive retrieval of cells from the chiral films using circularly polarized light played crucial role in regenerative medicine and tissue engineering. Compared with the conventional procedures like digestive enzymes, the irreversibly damage of the cells was almost negligible (**Figure S36, S40**).

We detached cells from chiral films and transferred them to another L- and D-Pen-NP film with addition of retinoic acid, the cells differentiated well after 6 days as shown in **Figure R4 and R5**. The results showed that the detached cell populations were stable and remain the strong viability to differentiate.

The technology is designed to promote a cell response in type and numbers perhaps, and be able to easily remove them for other use. This is useful technique to have when manipulating cells for therapy and tissue engineering in the culture system.

The experimental results and related discussion were added into main text and SI.

Figure R4 (Figure S45). MTT assay of cells detached by NIR light and transferred to another L- (A) and (B) D- Pen-NP film.

Figure R5 (Figure S46). Confocal images of cells detached by NIR light and transferred to another L- (A) and (B) D-Pen-NP film then differentiated (with addition of retinoic acid). Red, actin; green, vinculin; blue, nucleus. Scale bars were 20 μ m.

The detached cell populations were stable and remain the strong viability to differentiate (**Figure S45-46**).

My other comments:

Comments 2-3: The discussion on the plasmonic chirogenesis in page 5 line 109-115 is not clear. Authors should explain this part better.

Reply: Thank you for your suggestion. The plasmonic chirogenesis was explained as follows (See S15-16).

1) Au NPs (47 ± 5 nm) with an aspect ratio of 1.4 ± 0.2 (**Figure S14**) appear to be ellipsoids rather than spheres, so the balance between electrostatic repulsion and Van der Waals forces contributes to the dihedral angle between adjacent NPs in the monolayer film; 2) because Pen has a positive stereo-configuration, the spatial conformation of the NPs was reconfigured, enhancing the chiral configuration; and 3) the plasmonic hotspot enhanced the near-field dipolar Pen–gold interaction. According to electric field simulation (**Figure R6**), with the gap of 1.5 nm, the electric field 43.6 V/m that indicates 10.7 times enhancement compared to single nanoparticle (4.06 V/m). Additionally, CD and UV-Vis spectra of the chiral NP films have been simulated as **Figure R7** displayed, which was in consistent with the experimental results.

Figure R6 (Figure S15). Electric field simulation of Au NP-film and Au NP. A, side view of Au-NP film; B, top view of Au-NP film; C, Au NP. The beam for excitation was set in the z-axis direction at wavelength of 800 nm, the electric field was designed in the y-axis direction with initial values of 1 V/ m.

Figure R7 (Figure S16). Simulated CD (A) and UV-Vis (B) spectra of chiral NP film.

3) The plasmonic hotspot enhanced the near-field dipolar Pen-gold interaction. According to electric field simulation (Figure S15), with the gap of 1.5 nm, the electric field 43.6 V/m that indicates 10.7 times enhancement compared to single nanoparticle (4.06 V/m). Additionally, CD and UV-Vis spectra of the chiral NP films

have been simulated as Figure S16 displayed, which was in consistent with the experimental results.

Comments 2-4: Authors used fibronectin (I assume as a representative example of protein) to investigate the stereospecific interaction between cells and enantiomeric NP films. Then they claimed that (page 7 line 138-140) “Therefore, the distinct cell morphologies and proliferation behaviors of cells on enantiomeric NP films might be attributable to the different signals released from the stereospecific interactions between FN and the chiral NP films (Figure 1B).” Why did they use fibronectin? And how do they know that fibronectin is the result of this observed effect because cell experiments were done in serum (FBS) containing media and in FBS there are many other proteins available as well.

Reply: Thank you for your question. We are very sorry that we didn't explain the background of fibronectin clearly in the original text.

Fibronectin was a protein that promotes cell adhesion⁶⁻⁷. Fibronectin plays a major role in cell adhesion, growth, migration, and differentiation, and it is important for processes such as wound healing and embryonic development⁸⁻⁹. Therefore, the fibronectin was selected as a representative example of protein to investigate the stereospecific interaction between cells and enantiomeric NP films.

To further investigate the stereospecific interaction between cells and enantiomeric NP films, the adsorption behavior of fibronectin (FN) (a well-known protein that promotes cell adhesion) on experimental substrates was tested using an enzyme-linked immunosorbent assay (ELISA) kit.⁶⁸

(68)Hubmacher, D.; Sabatier, L.; Annis, D. S.; Mosher, D. F.; Reinhardt, D. P., Homocysteine Modifies Structural and Functional Properties of Fibronectin and Interferes with the Fibronectin-Fibrillin-1 Interaction, *Biochemistry* **2011**, *50*, 5322.

Comments 2-5: Authors mentioned that (page 7 line 145-152) “The CD signals of the chiral NP film were also astonishingly enhanced after cell adhesion and proliferation ... of the film in the Vis-NIR region”. But they did not discuss what most likely reason is? For example they can discuss that cells has intrinsic chirality maybe this is the reason of the enhanced CD signals etc.

Reply: Thank you very much for your constructive comments. According to your nice suggestion, the discussion about the reason for CD signals enhancement after cells adhesion has been added in the revised version. Intrinsic cell chirality has been found in the left-right-polarized protrusion of neutrophil-like cells *in vitro*¹. Cells could form invariant chiral alignment depended on the cell phenotype and actin function, indicating that all cells are intrinsically chiral². The chiral cells-plasmon dipolar interaction and plasmonic hot-spot resulted in the enhancement of CD signals. CD signals of cell and Pen in ultraviolet region induced CD enhancement of NP film in visible and near-infrared region.

The enhanced CD signals might be ascribed to induction of the intrinsic chirality of cells.^{6, 69} Cells could form invariant chiral alignment depended on the cell phenotype and actin function, indicating that all cells are intrinsically chiral.⁷⁰ The chiral cells-plasmon dipolar interaction and plasmonic hot-spot resulted in the enhancement of CD signals.

(6) Taniguchi, K.; Maeda, R.; Ando, T.; Okumura, T.; Nakazawa, N.; Hatori, R.; Nakamura, M.; Hozumi, S.; Fujiwara, H.; Matsuno, K., Chirality in Planar Cell Shape Contributes to Left-Right Asymmetric Epithelial Morphogenesis, *Science* **2011**, 333, 339.

(69) Yamanaka, H.; Kondo, S., Rotating pigment cells exhibit an intrinsic chirality, *Genes to Cells* **2015**, 20, 29.

(70) Wan, L. Q.; Ronaldson, K.; Park, M.; Taylor, G.; Zhang, Y.; Gimble, J. M.; Vunjak-Novakovic, G., Micropatterned mammalian cells exhibit phenotype-specific left-right asymmetry, *Proc. Natl. Acad. Sci. U. S. A.* **2011**, 108, 12295.

Comments 2-6: Authors mentioned that (page 9, line 189-195) “The cells on the L-Pen-NP film retained a living rate exceeding 95%, whereas the viability of the cells on the D-Pen-NP film decreased to 40%–60% (Figure 5C, S36–37). Although the cells on PDMS also remained in a good state, the detachment effect was quite weak (Figure S38–39). A flow-cytometric analysis (Figure S40–43) was used to confirm the low apoptosis rate (6%–8%) of the cells on the L-PEN-NP film.” Again Authors should discuss more in detail the most likely reason of their observation (The result

indicated that the left-handed film might have a protective effect on cells against the NIR laser as special ECM). For example since more proteins adsorbed on the surface of L-enantiomer this may work as a protecting layer etc.

Reply: Many thanks for your excellent advice. As your suggestion, we prepared L- and D- Pen-NP film for protein adhesion experiments. After 1 h incubation of the films in cell medium (10% fetal bovine serum), N element was tested via X-ray photoelectron spectroscopy (XPS). As the result shown (**Figure R8**), the amounts of background N element prior to protein adsorption were very low and similar between L- and D- Pen-NP films. However, after protein adsorption, the amount of N element on L-Pen-NP film was higher than that on D-Pen-NP film. This result might provide solid support for the protective effect on cells against the NIR laser.

Figure R8 (Figure S44). XPS spectra of N element of Au NP films before and after protein (cell medium with 10% fetal bovine serum) adsorption.

L- and D-Pen-NP films were prepared for protein adhesion experiments. After 1 h incubation of the films in cell medium (10% fetal bovine serum, FBS), N element was tested via X-ray photoelectron spectroscopy (XPS). As the result shown (Figure S44), the amounts of background N element prior to protein adsorption were very low and

similar between L- and D-Pen-NP films. However, after protein adsorption, the amount of N element on L-Pen-NP film was higher than that on D-Pen-NP film. This result indicated that more proteins adsorbed on the L-Pen-NP film might work as a protecting layer.

Comments 2-7: I think the reason of the use of NIR circularly polarized light (CPL) to improve the efficacy of light in cell detachment was not very well described even though this is the most innovative and interesting part of this study.

Reply: Thanks for your suggestion. We would like to explain more about the reason of using NIR circularly polarized light (CPL) to improve the efficacy of light in cell detachment.

Circular dichroism (CD) is the differential absorption of left- and right-handed circularly polarized light¹⁰⁻¹². The reason why LCP light led to the high detachment efficiency for cells on L-Pen-NP film was that LCP light activated a larger number of L-Pen-NPs than other light conditions, which increased the number of photoejected hot electrons of chiral film by adsorbing the related rotation of polarized light.¹³ Therefore, the chiral films absorbing CPL with high efficiency resulted in desirable cell detachment.

The reason why LCP light led to the high detachment efficiency for cells on L-Pen-NP film was that LCP light activated a larger number of L-Pen-NPs than other light conditions, which increased the number of photoejected hot electrons of chiral film by adsorbing the related rotation of polarized light.⁶⁷

(67)Hao, C.; Xu, L.; Ma, W.; Wu, X.; Wang, L.; Kuang, H.; Xu, C., Unusual Circularly Polarized Photocatalytic Activity in Nanogapped Gold-Silver Chiroplasmonic Nanostructures, *Adv. Funct. Mater.* **2015**, *25*, 5816.

Reviewer #3 (Remarks to the Author):

Comments 3-1: This paper introduces a nice platform innovation for cell-material interplay leading to increased proliferation, directed differentiation and for cell retrieval using the unique properties of surface chiral configuration and architectures. The study is significant in that it addresses an important and biologically relevant topic that has been overlooked in cell manipulation techniques and technology as well

as in the design of biomaterials. For me the study requires more breadth and depth to be accepted for publication in Nature Communications.

I do not find the application of the chiral films convincing enough to warrant publication in Nature Communications as they stand. The study is technically sound and accurate, but is a little too limited for the Nature Communications forum. There are large limitations in the cell studies. Principally it needs to go further in the process and analysis. The materials innovation is good and the materials work is high quality and tight.

Reply: Many thanks for your suggestion. The significance of this manuscript is in four aspects.

1. Monolayer chiral Au nanoparticle (NP) film with circular dichroism (CD) peaks at 550 nm and 775 nm was first fabricated for cell growth, differentiation and retrieval.
2. It was found that L-Pen-NP film could accelerate cell proliferation while D-Pen-NP film exhibited opposite effect, which could be ascribed to stereospecific interaction between fibronectin and chiral NP film.
3. The twisted spatial conformation of NP at nanoscale provided significant guidance in neurite outgrowth. Cells on L-Pen-NP film displaying dipolar differentiation, whereas multipolar neurite outgrowth occurred on D- Pen-NP film.
4. Due to the broad plasmon absorbance band in visible and near-infrared region, circularly polarized light (808 nm laser) was conducted for high efficiency non-invasive cell harvest.

To go further in the process and analysis of cell studies, we have performed further experiments as follows.

1. Besides the chiral penicillamine, we also applied other chiral molecules like L/D-cystine, L/D-phenylalanine to the preparation of chiral NP films, and similar results were obtained (**Figure R9**).
2. To evaluate the stability and ability to differentiate of the detached cells, we detached cells from chiral films using near-infrared circularly polarized light and transferred them to another L- and D-Pen-NP film with addition of retinoic acid. The cells differentiated well after 6 days as shown in **Figure R4, 5**. These results indicated that the detached cell populations were stable and remain the strong viability to differentiate.

In summary, the developed monolayer plasmonic NP films with strong chirality remarkably affected the cell adhesion, proliferation, and directional differentiation. Cells adhered on chiral film exhibited differentiation behaviors, which might have good utility in the repair of nerve damage. Importantly, this phenomenon could be attributed to chiral self-sorting between the chiral penicillamine modified film surface and cytomembrane proteins¹⁴⁻¹⁵. The investigation might help to understand the chirality selection of biosystems, and shed new insight into development of next-generation chiral biomaterials.

The related part was also wrapped into the main text.

Figure R9 (Figure S30). Confocal images of NG108-15 cells adhered on L-NP films and D-NP films functionalized with L/D-cystine (lane a) or L/D-phenylalanine (lane b) without (8 hr) and with addition of retinoic acid (RA, 6 days). (Red, actin; green, vinculin; blue, nucleus.) Scale bars were 20 μ m.

1. Adherent cells showed stretching when grown on the L-NP film (functionalized with L-cystine or L-phenylalanine), whereas the cells on the D-NP film (functionalized with D-cystine or D-phenylalanine) had a predominantly round morphology.
2. Cells on the L-NP film (functionalized with L-cystine or L-phenylalanine) displayed bipolar differentiation. In contrast, the cells on the D-NP film (functionalized with D-cystine or D-phenylalanine) showed multipolar neurite outgrowth.

Figure R4 (Figure S45). MTT assay of cells detached by NIR light and transferred to another L- (A) and (B) D-Pen-NP film.

The detached cells were almost in strong viability (**Figure R4**).

Figure R5 (Figure S46). Confocal images of cells detached by NIR light and transferred to another L- (A) and (B) D-Pen-NP film then differentiated (with addition of retinoic acid). Red, actin; green, vinculin; blue, nucleus. Scale bars were 20 μ m.

Besides the chiral penicillamine, we also applied other chiral molecules such as L/D-cystine, L/D-phenylalanine to the preparation of chiral NP films, and similar results were obtained (Figure S30).

Comments 3-2: This is an interesting study in parts, but it remains incohesive; meaning that I am unclear about its primary focus and the significance it has for the stated aims, which were stated as tissue engineering and cell manipulation. More explanation is needed on the problem that can be tackled with this technology. I would like to see more evidence of the cell growth and differentiation into structures with biotechnological value. I think the neurite morphology shape change on the chiral surfaces is really interesting. Cells that are grown on this platform are preferentially differentiated on the L-form surface and detached. Are the detached cell populations stable and do they retain their differentiated state?

Reply: Thanks for your nice question very much. Yes. It is.

We have conducted that experiment carefully and the result was displayed as below. After cells cultured on chiral films (with addition of retinoic acid) differentiated, the cells were irradiated with LCP (for L-Pen-NP film) or RCP (for D- Pen-NP film) for 5 min. Then, the detached differentiated cells were transferred to normal tissue culture plate. 24 h later, the morphology of cells were characterized with confocal image and the expression of N-myc oncoprotein was confirmed by western blot. As shown in **Figure R10** and **Figure R11**, the detached cells populations were stable and retain their differentiated state well. These results indicated that the chiral film could be used as a platform to guide cell differentiation and have potentials for repair of nerve damages. Also, compared with the conventional procedures like digestive enzymes, cell detachment realized by NIR light (especially circular polarized light) could almost avoid the irreversibly damage of the cells. The controlled and noninvasive retrieval of cells from responsive substrates would play a crucial part in regenerative medicine and tissue engineering.

Figure R10 (Figure S55). Confocal images of the differentiated cells transferred to normal tissue culture plate 24 h later (A, cells differentiated on L-Pen-NP film and then were transferred to normal tissue culture plate. B, cells differentiated on D-Pen-NP film and then were transferred to normal tissue culture plate.) (red, actin; green, vinculin; blue, nucleus). Scale bars were 20 μ m.

Figure R11 (Figure S56). Expression of N-myc protein in the differentiated cells and then transferred to normal tissue culture plate 24 h later (A, control. B, cells differentiated on D-Pen-NP film and then were transferred to normal tissue culture plate. C, cells differentiated on L-Pen-NP film and then were transferred to normal tissue culture plate.).

Besides the morphological differentiation, the expression of N-myc oncoprotein was proved to be decreased in NG108-15 cells. As shown in Figure 4B, cells differentiated on L-Pen-NP film exhibited lowest N-myc protein expression, which was consisting with the neurite outgrowth results. After cells differentiated on L- or D-Pen-NP film

and then were transferred to normal tissue culture plate, the expression of N-myc oncoprotein was confirmed by western blot as **Figure R11** shown. This result indicated and confirmed that the detached cells populations were stable and retains their differentiated state well.

In order to verify the cells grown on the developed platform were preferentially differentiated on the L-Pen-NP film and detached, we detached the cells using CPL and transferred them to normal tissue culture plate. The results revealed that the detached cells populations were stable and retain their differentiated state well (Figure S55-56).

Comments 3-3: The tuning effect with the plasmon film is novel and interesting too, but much more detail on its effect on cells, collective cell behaviour and applications are needed.

Reply: Thank you for your comments and suggestions. In this study, we designed enantiomorphous penicillamine functionalized monolayer Au NP films for cell interactions.

Firstly, cells could recognize and differentiate between the different chiral surfaces. Most adherent cells showed stretching when grown on the L-Pen-NP film, whereas the cells on the D-Pen-NP film had a predominantly round morphology. The aspect ratio of the cells on the L-Pen-NP film was approximately 3.3 times higher than that on the D-Pen-NP film. Furthermore, the quantity of cells adhered on the L-Pen-NP film was 2.2 times higher than that on the D-Pen-NP film. The distinct cell morphologies and proliferation behaviors of cells on enantiomeric NP films might be attributable to the different signals released from the stereospecific interactions between FN and the chiral NP films. It has been suggested that chiral molecules can affect proteins by non-covalent interaction, such as stereo-selective hydrophobic and hydrogen bonding effects. These effects may influence the interactions of cells with chiral surfaces as shown on the **Figure 1**.

Secondly, NG108-15 cell cultured on the enantiomeric NP films exhibited distinct differentiation behaviors with addition of retinoic acid. Cells on the L-Pen-NP film displayed bipolar differentiation, while cells on the D-Pen-NP film showed multipolar neurite outgrowth. This chiral platform might have good potentials in directional differentiation of stem cell for repair of nerve damages.

Thirdly, the developed chiral plasmonic film could be used as versatile platform for cell retrieval. The use of circular polarized light for cell retrieval improved the detachment efficiency and reduced the invasion to cell with nearly complete viability of the detached cells. Moreover the plasmonic film could also be applied to kinds of cell lines. This technique has a significantly potential in cell biology and biomedicine.

Comments 3-4: The abstract. The first sentence is correct but it is also important in biology and medicine too.

Reply: Thanks very much for your kind suggestion. Designing chiral materials to manipulate the biological activities of cells has been an important area not only in chemistry and material science, but also in biology and medicine. And we have revised the first sentence of the abstract as your advice.

Comments 3-5: I am confused at the start on the specific application of the chiral film? You should state clearly that the technology is designed to promote a cell response in type and numbers perhaps, and be able to easily remove them for other use. I am not clear on the removal mechanism that you have invented.

Reply: Thank you for your suggestion and question. We have stated that this technology is designed to promote a cell response in type and numbers perhaps, and be able to easily remove them for other use in the main text already. Please see **Figure 3 B and C**, page 4 line12-13.

The removal mechanism was the photothermal effect caused by the strong plasmon absorption of the Au NP film in the 700 – 900 nm range¹⁶⁻¹⁷. Therefore, a laser at 808 nm was chosen for the near-infrared (NIR)-triggered detachment of the living cells. Illumination with a laser at 808 nm for cell retrieval improved the detachment efficiency and reduced the invasion to cell with nearly complete viability of the detached cells. (**Figure 5C**, **Figure S40–43**).

To answer this question, we prepared L- and D-Pen-NP film for protein adhesion experiments. After 1 h incubation of the films in cell medium (10% fetal bovine serum), N element was tested via X-ray photoelectron spectroscopy (XPS). As the result shown (**Figure R8**), the amounts of background N element prior to protein adsorption were very low and similar between L- and D-Pen-NP films. However, after protein adsorption, the amount of N element on L-Pen-NP film was higher than that

on D-Pen-NP film. This result might provide solid support for the protective effect on cells against the NIR laser.

Figure R8 (Figure S44). XPS spectra of N element of Au NP films before and after protein (cell medium with 10% fetal bovine serum) adsorption.

Comments 3-6: I liked the idea that the chiral surface facilitated the detachment of cells using the light irradiation method. That is useful technique to have when manipulating cells for therapy and tissue engineering in the culture laboratory. Maybe you could emphasize this. Use your platform for driving cells into a fate and populations that form volumes or functional structures that can be transferred to another culture system or be implanted?

Reply: Thanks for your comments and question very much.

We have emphasized that this is useful technique to have when manipulating cells for therapy and tissue engineering in the culture laboratory in the main text.

To answer your question,

Firstly, the expression of N-myc oncoprotein was proved to be decreased in NG108-15 cells. As shown in Figure 4B, cells differentiated on L-Pen-NP film exhibited lowest

N-myc protein expression, which was consistent with the neurite outgrowth results (Figure 4C, D).

Secondly, cells differentiated into morphologies were transferred to normal tissue culture plate. 24 h later, the morphology of cells were characterized with confocal image and the expression of N-myc oncoprotein was confirmed by western blot. As shown in **Figure R10** and **Figure R11**, the detached cells populations were stable and retain their differentiated state well. These results indicated that the chiral film could be used as a platform to guide cell differentiation and have potentials for repair of nerve damages and tissue engineering.

Figure R10 (Figure S55). Confocal images of the differentiated cells transferred to normal tissue culture plate 24 h later (A, cells differentiated on L-Pen-NP film and then were transferred to normal tissue culture plate. B, cells differentiated on D-Pen-NP film and then were transferred to normal tissue culture plate.) (red, actin; green, vinculin; blue, nucleus). Scale bars were 20 μ m.

Figure R11 (Figure S56). Expression of N-myc protein in the differentiated cells and then transferred to normal tissue culture plate 24 h later (A, control. B, cells differentiated on D-Pen-NP film and then were transferred to normal tissue culture plate. C, cells differentiated on L-Pen-NP film and then were transferred to normal tissue culture plate.).

Comments 3-7: What is the ultimate destination for this technology in biomedicine? Please add specific applications in this regard.

Reply: Many thanks for your suggestion. The findings of the investigation have shown that the growth, differentiation and polarization of nerve cells are powerfully influenced by cell and ECM interaction. The nanotopographical features and surface molecules of ECM could play significant role in controlling the polarity, directional growth and neuronal differentiation. Cells on the L-Pen-NP film displayed bipolar differentiation, whereas multipolar neurite outgrowth occurred on the D-Pen-NP film. Also, cells developed into a fate and populations that form volumes or functional structures could be transferred to another culture system through light triggered detachment without damage. Looking to the future, this platform might provide opportunity to control the axonal alignment and growth of neural stem cell derived neurons for the development of more effective treatment for spinal cord injuries. This technique might facilitate the cell culture and cell engineering, including in the therapeutic use of stem cells. In-depth studies may inspire the design of artificial prosthetic devices and utility of these biomaterials to recapitulate biological or natural phenomena.

Comments 3-8: I don't like the last sentence as it is too general. Many other studies have demonstrated the utility of biomaterials to recapitulate biological/ natural phenomena. Also I did not get the sense that this study “clarified chiral selection” more it added to the body of evidence showing that chirality influences cell behaviour, in this case neurite growth. In fact the cell behaviours were altered by the combination of chiral patterned surface and chemical inductor.

Reply: Thanks for your suggestions very much. According to your suggestion, the

last sentence has been revised as “In-depth studies may inspire the design of artificial prosthetic devices and utility of biomaterials to recapitulate biological or natural phenomena.” The “clarified chiral selection” is attributed to chiral self-sorting between the chiral molecules modified film surface and cytomembrane proteins.¹⁴⁻¹⁵

We added the related experiments to answer your questions.

As fibronectin (FN) was a well-known protein that promotes cell adhesion, we studied the FN adsorption on the substrates to investigate the stereospecific interaction between cells and films. As shown in **Figure R12**, the quantity of FN adsorbed onto the L-Pen-NP film was 1.74 times greater than that on the D-Pen-NP film within the same culture condition. Therefore, the chirality of the NP film was mainly recognized by the stereospecific interaction between FN (on the cell surface) and the chiral films. Furthermore, the isothermal titration calorimetry (ITC) analysis (**Figure R13**) was demonstrated that the K_a between L-Pen-NP and FN (1.566×10^5) was almost 15 times of that between D-Pen-NP and FN (1.101×10^4). These data showed that the chirality influenced cell behavior in the neurite growth.

Figure R12 (Figure S22). Time-dependent adsorption of FN on PDMS, NP film, L-Pen-NP film and D-Pen-NP film.

For the FN adsorption test, 1 mL of FN solution (1000 pg/mL) was added to each well plated with substrate ($1 \times 1 \text{ cm}^2$) in a 24-well culture plate, and the plate was

incubated at 37 °C in an atmosphere containing 5% CO₂. After the FN had adsorbed, the FN left in the supernatant was quantified with an enzyme-linked immunosorbent assay kit (Wuhan boster biological technology).

Figure R13. Isothermal titration calorimetry (ITC) data for chiral interactions and their fitting with thermodynamic models. Corrected heat rate of FN into L-Pen-NP (A) and FN into D-Pen-NP (B). Modeling of FN into L-Pen-NP (C) and D-Pen-NP (D). FN into water was set as a control experiment.

Comments 3-9: The Introduction needs improving. It needs to have more rigid cohesion and more breadth. I would particularly be interested to read a stronger rationale for designing artificial ECM with chirality. I am with you on its significance as all biological systems have chirality.

Reply: Thanks a lot for your comments. We have revised the introduction according to your suggestion.

Here we reported that monolayer film fabricated with plasmonic nanoparticles with strong chirality was used as an ECM for first time. As well as the chirality of the molecules in the ultraviolet region, the chiral NP films with highly intense circular dichroism (CD) peaks at 550 nm and 775 nm play a vital role in cell-ECM interactions.

Comments 3-10: Maybe you could identify the chiral features of ECM that have been reported to support your statements on the importance of ECM-cell interactions. I would like to read about the studies investigating this complementarity.

Please include, albeit in other ways, that materials have been designed and fabricated to add chiral features and signatures. There is an important set of literature on chiral supramolecular biomaterials.

Reply: Thanks a lot for your comments and suggestion.

Supramolecular biomaterials with reversible, highly tunable and dynamic fashion provide a diverse toolbox that could help to address important unmet biomedical needs.¹⁸⁻¹⁹ Bioactive supramolecular materials formed through incorporation of chiral groups into building blocks showed good potential as biocompatible scaffolds.^{17, 20-21} The introduction has been revised and literature on chiral supramolecular biomaterials has been added in the revised introduction.

Bioactive supramolecular materials formed through incorporation of chiral groups into building blocks showed good potential as biocompatible scaffolds.³⁸⁻⁴⁰

(38) Dong, R.; Zhou, Y.; Huang, X.; Zhu, X.; Lu, Y.; Shen, J., Functional Supramolecular Polymers for Biomedical Applications, *Adv. Mater.* **2015**, *27*, 498.

(39) Marchesan, S.; Styan, K. E.; Easton, C. D.; Waddington, L.; Vargiu, A. V., Higher and lower supramolecular orders for the design of self-assembled heterochiral tripeptide hydrogel biomaterials, *J. Mater. Chem. B* **2015**, *3*, 8123.

(40) Marchesan, S.; Easton, C. D.; Styan, K. E.; Waddington, L. J.; Kushkaki, F.; Goodall, L.; McLean, K. M.; Forsythe, J. S.; Hartley, P. G., Chirality effects at each amino acid position on tripeptide self-assembly into hydrogel biomaterials, *Nanoscale* **2014**, *6*, 5172.

Comments 3-11: Detail to add on bland statement: “These studies have shown that cells can interact with chiral surface molecules and display different behaviors on enantiomorphous surfaces”

Reply: Thanks for your suggestion.

The detail of the statement has been added in the main text as follows.

The cells were found to adhere on the D-surface much less than that on the L-surface.⁴²⁻⁴⁴

(42)Hanein, D.; Geiger, B.; Addadi, L., Differential Adhesion of Cells to Enantiochiral Crystals-Surfaces, *Science* **1994**, *263*, 1413.

(43)Liu, G. F.; Zhang, D.; Feng, C. L., Control of Three-Dimensional Cell Adhesion by the Chirality of Nanofibers in Hydrogels, *Angew. Chem. Int. Ed.* **2014**, *53*, 7789.

(44)Sun, T.; Han, D.; Rhemann, K.; Chi, L.; Fuchs, H., Stereospecific interaction between immune cells and chiral surfaces, *J. Am. Chem. Soc.* **2007**, *129*, 1496.

Comments 3-12: Make a case as to why nanoparticles attached to a 2D platform would be used as an artificial ECM system?

Reply: Thanks for your constructive suggestions.

Designing biomaterials with distinct merits as platform to recapitulate the native extracellular matrices (ECM) in tissue regeneration and injury recovery has drawn increasing appreciation in material science, biology and medicine.²²⁻²³ Surface topography at the nanoscale not only determined the early in vitro neurite development rate, but also could accelerate neurite outgrowth and polarization of neurons.²⁴⁻²⁶

To guide for neuron cells development, the nanoparticles attached 2D platform could be used for cell culture and efficient detachment. Qu's group used a spiropyran-conjugated upconversion nanoparticles on glass as a photoswitch to control cell detachment.¹⁷ And Liz-Marzan and coworkers designed a gold nanoparticles on glass for cell growth and detached the cells with near-infrared (NIR) light.¹⁶

Therefore, functionalized 2D nanomaterials have been used as a model system of ECM.

Comments 3-13: It would be nice to know whether the neurites from synaptic connections and can be proved to transmit electrical signals while attached onto this special surface.

Reply: Thanks for your comments. In this research, we focus on Au NPs with minimal toxicity¹⁴ and highly conductivity²⁷ as biomaterial suitable for neural interface. Currently, most research on nanomaterials for neural interface is mainly

focused on carbon-based material.²⁸⁻²⁹ And it has been demonstrated that electrical signaling in neural networks on graphene films and carbon nanotube materials could be enhanced.³⁰⁻³¹ Besides the higher cell density and accelerated neurite outgrowth, the inherent electrical conductivity of the substrates was another significant factor for the increasing efficacy of neural signal transmission and electrical activity.³²⁻³⁴ Compared to carbon-based materials, conductivity of macroscale composite materials made from Au NPs is higher.³⁴⁻³⁵ In addition, the proposed L-penicillamine functionalized Au NP film exhibited excellent promotion for cell adhesion, proliferation (**Figure 3C**) and neurite outgrowth (**Figure 4**). On the one hand, the tight connection between the neurites proved that the neurons on the chiral Au NP films could form structural connections. On the other hand, the high electrical conductivity of the films guarantee electrical coupling between the cells and substrate. This developed chiral Au NP films might be a good candidate for neuronal electrical signal transmitting.

Comments 3-14: I would suggest a clearer description of the aims of your study. The introduction is a confusing preface to this study. I was led to think at the start that the aim was to fabricate ECM mimics for cell biocompatibility. Then we moved towards a technical device for cell retrieval. The results are interesting. So, we now know that your chiral surfaces promote neurite cells in favourable ways.

Reply: Thanks a lot for your nice suggestion. This study designed chiral nanoparticle films to promote cell response in type and numbers, and easily retrieval the cells without damage. We rewrote the aims of our study in the revised text according to your suggestion.

Comments 3-15: Use of headings in the script will be helpful.

Reply: Thanks a lot. We have added headings in the revised version.

Comments 3-16: You state that the study could lead onto better understanding of chiral selectivity, so can you adapt your fabrication to create lots of chiral variants? Can you replicate the chiral signatures seen in nature onto your platform? How versatile is the platform to do this and work for biotechnology, cell therapy requirements? I think you have a nice platform technology but more work is needed to show its good potential. For me the studies layed down do not do that.

Reply: Thanks for your comments.

To create chiral variants, we applied other typical amino acids (e.g. L/D-cystine, L/D-phenylalanine) to prepare chiral NP films. Cells could also recognize and differentiate between the enantiomeric films (**Figure R9**). The cell adhesion was enhanced by the L surface. This could be ascribed to the preferred adsorption of proteins (cell medium with 10% fetal bovine serum, **Figure R8**) and chiral self-sorting. The cell tension (relative to cell shape) and spatial conformation of the NPs affected the forthcoming differentiation (neurite outgrowth). The study might help to the development of platform for directional differentiation of stem cells for damaged nerve repair.

Chirality is one of the most significant biochemical signatures of life.³⁶ Biomolecules in nature are usually chiral and biosystem ordinarily exhibit high chiral preference for one specific enantiomer (L-amino acids, D-sugars, etc.).^{3, 37} Intrinsic cell chirality has been found in the left-right polarized protrusion of neutrophil-like cells in vitro.¹ Cells could form invariant chiral alignment depended on the cell phenotype and actin function, indicating that all cells are intrinsically chiral.² And previous studies proved that molecules or nanoparticles of the same chirality are attracted to each other more strongly than with opposing chirality.^{15, 38} This interesting phenomenon is named chiral self-sorting. In this work, cells preferred to adhere and proliferate on L-Pen-NP film as opposed to D-Pen-NP film. This situation might also be exemplification of chiral self-sorting.

The fabricated monolayer plasmonic NP films with strong chirality noticeably possess the ability to affect cell adhesion, proliferation, and directional differentiation. Looking to the future, this platform might provide opportunity to control the axonal alignment and growth of neural stem cell derived neurons for the development of more effective treatment for spinal cord injuries.

Figure R9 (Figure S30). Confocal images of NG108-15 cells adhered on L-NP films and D-NP films functionalized with L/D-cystine (lane a) or L/D-phenylalanine (lane b) without (8 hr) and with addition of retinoic acid (RA, 6 days). (Red, actin; green, vinculin; blue, nucleus.) Scale bars were 20 μ m.

Figure R8 (Figure S44). XPS spectra of N element of Au NP films before and after protein (cell medium with 10% fetal bovine serum) adsorption.

Comments 3-18: It would be important to extend the cell studies beyond 5 minutes to realise the potential of the surfaces on proliferation, differentiation responses. The timing should be extended to determine that the cells are stable, stably attached and whether they process the new environment and develop into overt tissue structures, for instance.

Reply: Thanks for your comments and suggestion.

We did the cell detachment experiment and extend the light action time to 10 minutes. The results of the cell detachment are displayed in **Figure R14-16**. The function of near-infrared light in this experiment was photothermal effect to detach cells from the substrates. The detached cells (L-Pen-NP film, LCP, 10min and D-Pen-NP film, RCP, 10min) were stable and could be transferred to another system and develop very well (**Figure R17**).

Figure R14. Confocal images of NG108-15 cell grown on L-Pen-NP film upon polarized light irradiation (808nm laser, 150 mW/cm². LCP: left circularly polarized

light. LP: linearly polarized light. RCP: right circularly polarized light.). Scale bar, 20 μm .

Figure R15. Confocal images of NG108-15 cell grown on D-Pen-NP film upon polarized light irradiation (808nm laser, 150 mW/cm^2 . LCP: left circularly polarized light. LP: linearly polarized light. RCP: right circularly polarized light.). Scale bar, 20 μm .

Figure R16. Cell detachment rates upon laser irradiation (808nm laser, 150 mW/cm², 10 min).

Figure R17. Confocal images of NG108-15 cells detached from L-Pen-NP film (LCP, 10min) and D-Pen-NP film (RCP, 10min) and cultured on tissue culture plate with addition of retinoic acid (6 days).

Comments 3-18: You could have used more display items of up to 10 to increase the depth and breadth of the study material.

Reply: Thanks a lot for your suggestion. 6 items have been used. The others were added to SI.

Comments 3-19: I would like to see more highly resolved observation of the cell interaction mechanics with the chiral architecture and Fn associations and its conformation. You state that the film is an ECM.

Reply: Thanks for your comments.

In our text, cells (NG108-15, HeLa and PCS-460-010) cultured on the L-Pen-NP film showed high aspect ratio, whereas most cells on D-Pen-NP film had a round shape. The L-Pen-NP film could enhance the cell adhesion and proliferation, and D-Pen-NP film has the opposite effect (**Figure R18**). The different cell behaviors were mainly regulated by the chiral NP films. As fibronectin (FN) was a well-known protein that promotes cell adhesion, we studied the FN adsorption on the substrates to investigate the stereospecific interaction between cells and films. As shown in **Figure R12**, the quantity of FN adsorbed onto the L-Pen-NP film was 1.74 times greater than that on the D-Pen-NP film within the same culture condition. Therefore, the chirality of the NP film was mainly recognized by the stereospecific interaction between FN (on the cell surface) and the chiral films. Furthermore, the isothermal titration calorimetry (ITC) analysis (**Figure R13**) demonstrated that the K_a between L-Pen-NP and FN (1.566×10^5) was almost 15 times of that between D-Pen-NP and FN (1.101×10^4).

Figure R18. Confocal images (high magnification) of NG108-15 cell grown on L-Pen-NP film (A) and D-Pen-NP film (B).

Cells on L-Pen-NP film showed stretching, whereas cells on the D-Pen-NP film had a predominantly round morphology.

Figure R12 (Figure S22). Time-dependent adsorption of FN on PDMS, NP film, L-Pen-NP film and D-Pen-NP film.

For the FN adsorption test, 1 mL of FN solution (1000 pg/mL) was added to each well plated with substrate ($1 \times 1 \text{ cm}^2$) in a 24-well culture plate, and the plate was incubated at 37 °C in an atmosphere containing 5% CO₂. After the FN had adsorbed, the FN left in the supernatant was quantified with an enzyme-linked immunosorbent assay kit (Wuhan boster biological technology).

Figure R13. Isothermal titration calorimetry (ITC) data for chiral interactions and their fitting with thermodynamic models. Corrected heat rate of FN into L-Pen-NP (A) and FN into D-Pen-NP (B). Modeling of FN into L-Pen-NP (C) and D-Pen-NP (D). FN into water was set as a control experiment.

Comments 3-20: More evidence is needed to justify the claim that the nanosurfaces can be used to “clarify the chirality selection of biosystems”.

Reply: The “clarified chiral selection” is attributed to chiral self-sorting between the chiral molecules modified film surface and cytomembrane proteins.¹⁴⁻¹⁵

Similar chiral self-sorting has also been observed for fibrous protein hydrogels³⁹ and supramolecular chiral columns³⁸.

Ding’s group previously investigated the adhesion and differentiation of stem cells on gold coating (glass sputtered with gold) modified with L- or D-cysteine,⁴⁰ and Seda Kehr et al. reported cell interactions with a chiral-amino-acid-functionalized zeolite nanoparticle (NP) monolayer.⁴¹ These studies have shown that cells can interact with chiral surface molecules and display different behaviors on enantiomorphous surfaces. The cells were found to adhere on the D-surface much less than that on the L-surface.

We also added the related experiments to answer your questions.

As fibronectin (FN) was a well-known protein that promotes cell adhesion, we studied the FN adsorption on the substrates to investigate the stereospecific interaction

between cells and films. As shown in **Figure R12**, the quantity of FN adsorbed onto the L-Pen-NP film was 1.74 times greater than that on the D-Pen-NP film within the same culture condition. Therefore, the chirality of the NP film was mainly recognized by the stereospecific interaction between FN (on the cell surface) and the chiral films. Furthermore, the isothermal titration calorimetry (ITC) analysis (**Figure R13**) demonstrated that the K_a between L-Pen-NP and FN (1.566×10^5) was almost 15 times of that between D-Pen-NP and FN (1.101×10^4).

Figure R12 (Figure S22). Time-dependent adsorption of FN on PDMS, NP film, L-Pen-NP film and D-Pen-NP film.

For the FN adsorption test, 1 mL of FN solution (1000 pg/mL) was added to each well plated with substrate ($1 \times 1 \text{ cm}^2$) in a 24-well culture plate, and the plate was incubated at 37 °C in an atmosphere containing 5% CO₂. After the FN had adsorbed, the FN left in the supernatant was quantified with an enzyme-linked immunosorbent assay kit (Wuhan boster biological technology).

Figure R13. Isothermal titration calorimetry (ITC) data for chiral interactions and their fitting with thermodynamic models. Corrected heat rate of FN into L-Pen-NP (A) and FN into D-Pen-NP (B). Modeling of FN into L-Pen-NP (C) and D-Pen-NP (D). FN into water was set as a control experiment.

We also rewrote this sentence.

These findings will facilitate the development of cell culture in biomedical application and could help to understand the natural homochirality.

Reviewer #4 (Remarks to the Author):

Comments 4-1: The application of chiral surfaces is quite important in biomedical treatment and clinical detection. This manuscript prepared chiral penicillamine-modified nanoparticle coating PDMS films to control cell adhesion, differentiation and detachment. The results are very interesting. However, some problems in the writing, experiment and conclusion make it not well understood. Resubmission after more careful researches is suggested.

Reply: We thank the reviewer for their constructive feedback, and believe that the ensuing additions have strengthened the manuscript. According to your suggestions, we have thoroughly revised our manuscript to make extensive textual changes throughout the text of all the files to cause more easy understanding and also added the new data to the revised version that address you and other Reviewers' concerns.

Comments 4-2: There are a series of materials. However, they were named not systematically and given the definition from the beginning. For example, L-Pen-NP film and D-Pen-NP film in the manuscript had been also named as L-Pen NP film and D-PEN NP film. While the methods for L-/D-Au coating and L-/D-PDMS were not given. By the way, it is not clear if L-/D-PEN were synthesized by the authors or from any commercial product.

Reply: Thank you very much for your kindly remind and we are sorry of our careless to cause the reviewing difficulty.

We have completely proofread the abbreviation of those given materials in the manuscript, such as L-NP film and L-Pen-NP film united for L-Pen-NP film, D-NP film and D-Pen-NP film united for D-Pen-NP film, respectively. And the detailed methods of L/D-Pen-Au coating and L/D-Pen-PDMS were added into the Supplementary Materials. Penicillamine enantiomers (denoted as **L/D-Pen** in this manuscript) were purchased from Sigma-Aldrich, which has been added into the Supplementary Materials. The related revised version in the Supplementary Materials was as follows:

L-penicillamine and D-penicillamine (denoted as L-Pen and D-Pen, respectively) were purchased from Sigma-Aldrich.

Preparation of polydimethylsiloxane film (abbreviated as PDMS)

The PDMS mixture of base and cross-linker (Dow Corning Sylgard 184; the weight ratio of base to cross linker was 10:1) was stirred at least for 20 min, degased in vacuum for 10 min to get rid of bubbles on a clean glass at room temperature,

solidified at 70 °C for 2 h, and carefully peeled off from the glass to get PDMS thin film. It was then immersed in 3-aminopropyltriethoxysilane (APTES) solution (30 μ L APTES in 30 ml ethanol) for 1 h, then washed with ethanol and dried in the oven at 60 °C for 1 h.

Preparation of L/D- Pen-PDMS

To prepare L- or D-PDMS, 50 mM L- or D-Pen enantiomer dissolved in pH 7.4 phosphate buffer (PB) containing 0.5 mM 1-(3-dimethyl-aminopropyl)-3-ethylcarbodiimide hydrochloride (> 98%, Sigma Aldrich) and 0.05 mM N-hydroxysuccinimide (> 98%, Sigma Aldrich) for 6 h activation. After that, aminated PDMS was added into the above mixture, and then incubated for 4 h with gentle shaking at 60 °C to form the film. Finally, the modified PDMS film was washed with pH 7.4 PB for 3 times to remove the physically adsorbed penicillamine enantiomer.

Preparation of L/D-Pen-Au coating

PDMS film was sputtered with gold (expressed as Au coating) using a sputter coater. After that, the Au coatings were incubated with 50 mM L- or D-Pen enantiomer at a 60 °C water bath for 2 h. After that, the Au coatings were rinsed with pH 7.4 PB for 3 times to remove the un-conjugated Pen enantiomer. And, we named the Au coatings modified with L- or D-Pen enantiomer as the L- or D-Pen-Au coating, respectively.

Comments 4-3: The bond formed between PEN and AuNP should be Au-S. However, it looks like just physical adsorption from Figure 1B.

Reply: We are very sorry for our careless about the typographical error. We have revised the typo of **Figure 1B** in the main text as follows.

Figure 1. **A)** Schematic representation of NG108-15 cells grown and differentiated on chiral plasmonic films. **B)** Schematic presentation of the regulation of NG108-15 cell adhesion by stereospecific interaction between fibronectin and chiral NP film.

Comments 4-4: The Au nanoparticles were observed at the size of around 50 nm. As expected, after modified by PEN, their sizes would be a little bigger. However, no direct evidence was provided. One can read from Figure 2C and 2D with a conclusion about 30-40 nm, which is different from the results of Figure S1 and the sentence in Line 22 on Page 4.

Reply: We appreciate the nice comments you give us and we totally agree it. As you suggested, to slap measure the size of Pen modified NPs, we added the dynamic

light scattering (DLS) data to give us the adequate evidence to measure the size and size distribution of particles in liquids. As shown in **Figure R19**, the averaged hydrodynamic diameter of nanoparticles was 49 ± 3 nm, which is nice correlation with the dimensions obtained by the TEM. And, the averaged hydrodynamic size of PEN molecules functionalized NPs was 51 ± 4 nm, which is slightly larger than that of individual NPs owing to the formation of self-assembled monolayer of Pen molecules onto the surface of NPs. As for the size of individual NP in the atomic force microscopy (AFM) (**Figure 2C and 2D**), in-depth analysis of the vertical dimension in AFM NP images was carried out. As the “section command” in the nanoscope analysis software we used in the previous version, the surface profile was marked and the measured vertical distance between the two cursors was displayed, we found that the vertical distance was not accurately reflected the dimension of Pen-NP. Thus, we expanded the region between two cursors to the blank boundary to another, and we found that the average vertical distance of the upper end and the lower end of the measuring area in z axis was 14 ± 2 nm and -30 ± 3 nm, respectively. Similarly to the size measured by TEM, SEM and DLS, the size of the particles was 44 ± 5 nm. We have added more detailed analysis of the size measured by DLS (**Figure R19**) and AFM (**Figure R20**) in the revised version.

Figure R19 (Figure S2). Dynamic light scattering (DLS) size of individual NPs and L-Pen molecules modified NPs.

Figure R20 (Figure S7). Vertical distance of Au NP film.

Comments 4-5: The property of the films should be not just due to their chirality, but also the modified amount of L-/D-PEN. However, there is no result provided. To evaluate their effects on cell behaviors, they should be controlled at the same levels on the films. More importantly, different levels of L-/D-PEN on the surfaces can give the significant effects and trends on cell adhesion and differentiation.

Reply: We agree with the reviewer that the modified amount of L-/D-PEN could affect the property of chiral NP films to further cause the discriminating trends on cell adhesion and differentiation. Based on the reviewer's comments, X-ray photoelectron spectroscopy (XPS) characterization was carried out to evaluate the amount of L- or

D-Pen grafted to the Au NP film. From the narrow spectra of Au and S (**Figure R21** and **Figure R22**), the L-Pen-NP film and D-Pen-NP film exhibited similar amounts of the S and Au elements, indicating the similar amounts of L-Pen and D-Pen grafted onto the Au NP films.

The Au NP film in the main text was prepared with Au NP functionalized in 50 mM Pen solution. A higher concentration of Pen functionalization could influence the preparation of the NP film, bringing difficulty in film formation. Therefore, to evaluate the effects of different levels of L-/D-Pen (functionalized on NP film) on cell adhesion and differentiation, chiral films prepared by 20 mM, 40 mM, and 50 mM Pen functionalized NPs were used for cell culture experiments. As **Figure R22** shown, the amounts of S element of the three groups were distinct. As for cell adhesion and differentiation, there are clear trends. The more the Pen functionalized on NP film, the more notable the effect on cell stretch and neurite outgrowth (**Figure R23**). The average aspect ratio of cells adhered on L-Pen-NP film increased with the increase of amount of L-Pen (**Figure R24**). On the contrary, cells adhered on D-Pen-NP film exhibited smaller sticking area when the amounts of D-Pen increased (**Figure R23 and Figure R24**). And the average length of neurites increased with the increase of Pen amounts (**Figure R25**) after RA was added.

X-ray photoelectron spectroscopy characterization was carried out to confirm the similar amounts of L-Pen and D-Pen grafted onto the Au NP films (**Figure S8**).

Figure R21 (Figure S8). X-ray photoelectron spectroscopy of (A) Au and (B) S element of the L/D-Pen modified Au NP films.

Figure R22. XPS spectra of S element of the different amounts Pen modified Au NP films

Figure R23. Confocal images of NG108-15 cells adhered on L-Pen-NP films and D-Pen-NP films functionalized with 20 mM (lane a) or 40 mM phenylalanine (lane b) without (8 hr) and with addition of retinoic acid (RA, 6 days). (Red, actin; green, vinculin; blue, nucleus.) Scale bars were 20 μ m.

Figure R24. Aspect ratio of NG 108-15 cells cultured on L- or D-Pen-NP film modified with different amounts of L/D-Pen.

Figure R25. Mean lengths of neurites of NG108-15 cells differentiated on chiral films modified with different amounts of Pen.

Comments 4-6: Moreover, although FN adsorption increased on L-Pen-NP film, it couldn't give the explanation of the cellular morphology and neural differentiation. The mechanism should be investigated more carefully.

Reply: Thanks a lot for your suggestion.

These unique differentiation behaviors might be explained as follows. First, the Pen on the surface of the NP film can regulate multiple cellular functions.⁴² Second, the stereospecific interaction between FN and the chiral NP films caused the distinct cell morphologies on the D- and D-Pen-NP films with different aspect ratio (shape) shown in Figure 3b, which contributed to vastly different neurite outgrowth after the addition of RA.⁴⁻⁵ Third, the twisted spatial conformation of the NPs at the nanoscale provided significant guidance for neurite outgrowth,⁴³⁻⁴⁴ developing the characteristic functions of the cells. Besides the morphological differentiation, the expression of N-myc oncprotein was proved to be decreased in NG108-15 cells. As shown in **Figure 4B**, cells differentiated on L-Pen-NP film exhibited lowest N-myc protein expression, which was consistent with the neurite outgrowth results. The experimental data of distinct cells differentiation on the chiral films confirmed that the cells interacted with the chiral plasmonic NP film and responded to mechanical stimuli by converting them into biochemical signals, which were probably transmitted through integrin.⁴⁵ And the developed L-Pen-NP film might have great potential utility in the repair of sensory damage as bipolar cells could be specialized for the transmission of special senses, such as smell, sight, and hearing.

Second, the stereospecific interaction between FN and the chiral NP films caused the distinct cell morphologies on the L- and D-Pen-NP films with different aspect ratio (shape) shown in Figure 3b, which contributed to vastly different neurite outgrowth after the addition of RA.⁷⁴⁻⁷⁵

(74) Peng, R.; Yao, X.; Ding, J., Effect of cell anisotropy on differentiation of stem cells on micropatterned surfaces through the controlled single cell adhesion, *Biomaterials* **2011**, *32*, 8048.

(75) Yao, X.; Peng, R.; Ding, J., Effects of aspect ratios of stem cells on lineage commitments with and without induction media, *Biomaterials* **2013**, *34*, 930.

Comments 4-7: The illumination at 808 nm will promote cell detachment. While the mechanism is due to high temperature treatment. From Figure S31, the temperature of L-Pen-NP film could reach to almost 60 degree after 5 min. It's really an extreme condition for cell survival. And why the other materials and conditions just reached a lower temperature after laser excitation but with a higher cell death? How can L-Pen protect the cells?

Reply: Thanks a lot for your questions.

To answer this question, the related protein adsorption experiments were carried out. And we found that L-Pen-NP film absorbed more protein than D-Pen-NP film within the same culture time in cell medium (**Figure R8**). The absorbed protein on the surface of L-Pen-NP film might have protection effect on the photothermal damage.

The other point could also be considered. As we can see in **Figure S31**, the temperature of L-Pen-NP film was almost 50 degree. However, cells almost detached from the NP film before 5 min and suspended in cell medium. The results were shown on the **Figure 5** and **Figure 6**.

Figure R8 (Figure S44). XPS spectra of N element of Au NP films before and after protein (cell medium with 10% fetal bovine serum) adsorption.

References

- (1) Xu, J.; Van Keymeulen, A.; Wakida, N. M.; Carlton, P.; Berns, M. W.; Bourne, H. R., Polarity reveals intrinsic cell chirality, *Proc. Natl. Acad. Sci. U. S. A.* **2007**, *104*, 9296.
- (2) Wan, L. Q.; Ronaldson, K.; Park, M.; Taylor, G.; Zhang, Y.; Gimble, J. M.; Vunjak-Novakovic, G., Micropatterned mammalian cells exhibit phenotype-specific left-right asymmetry, *Proc. Natl. Acad. Sci. U. S. A.* **2011**, *108*, 12295.
- (3) Taniguchi, K.; Maeda, R.; Ando, T.; Okumura, T.; Nakazawa, N.; Hatori, R.; Nakamura, M.; Hozumi, S.; Fujiwara, H.; Matsuno, K., Chirality in Planar Cell Shape Contributes to Left-Right Asymmetric Epithelial Morphogenesis, *Science* **2011**, *333*, 339.

- (4) Peng, R.; Yao, X.; Ding, J., Effect of cell anisotropy on differentiation of stem cells on micropatterned surfaces through the controlled single cell adhesion, *Biomaterials* **2011**, *32*, 8048.
- (5) Yao, X.; Peng, R.; Ding, J., Effects of aspect ratios of stem cells on lineage commitments with and without induction media, *Biomaterials* **2013**, *34*, 930.
- (6) Hubmacher, D.; Sabatier, L.; Annis, D. S.; Mosher, D. F.; Reinhardt, D. P., Homocysteine Modifies Structural and Functional Properties of Fibronectin and Interferes with the Fibronectin-Fibrillin-1 Interaction, *Biochemistry* **2011**, *50*, 5322.
- (7) Liu, G. F.; Zhang, D.; Feng, C. L., Control of Three-Dimensional Cell Adhesion by the Chirality of Nanofibers in Hydrogels, *Angew. Chem. Int. Ed.* **2014**, *53*, 7789.
- (8) Bharadwaj, M.; Strohmeyer, N.; Colo, G. P.; Helenius, J.; Beerenwinkel, N.; Schiller, H. B.; Faessler, R.; Mueller, D. J., alpha V-class integrins exert dual roles in alpha 5 beta 1 integrins to strengthen adhesion to fibronectin, *Nat. Comm.* **2017**, *8*: 14348.
- (9) Slapsak, U.; Salzano, G.; Amin, L.; Abskharon, R. N. N.; Ilc, G.; Zupancic, B.; Biljan, I.; Plavec, J.; Giachin, G.; Legname, G., The N Terminus of the Prion Protein Mediates Functional Interactions with the Neuronal Cell Adhesion Molecule (NCAM) Fibronectin Domain, *J. Biol. Chem.* **2016**, *291*, 21857.
- (10) Wu, X.; Xu, L.; Liu, L.; Ma, W.; Yin, H.; Kuang, H.; Wang, L.; Xu, C.; Kotov, N. A., Unexpected Chirality of Nanoparticle Dimers and Ultrasensitive Chiroplasmonic Bioanalysis, *J. Am. Chem. Soc.* **2013**, *135*, 18629.
- (11) Yan, W.; Xu, L.; Xu, C.; Ma, W.; Kuang, H.; Wang, L.; Kotov, N. A., Self-Assembly of Chiral Nanoparticle Pyramids with Strong R/S Optical Activity, *J. Am. Chem. Soc.* **2012**, *134*, 15114.
- (12) Ferre, A.; Handschin, C.; Dumergue, M.; Burgy, F.; Comby, A.; Descamps, D.; Fabre, B.; Garcia, G. A.; Geneaux, R.; Merceron, L.; Mevel, E.; Nahon, L.; Petit, S.; Pons, B.; Staedter, D.; Weber, S.; Ruchon, T.; Blanchet, V.; Mairesse, Y., A table-top ultrashort light source in the extreme ultraviolet for circular dichroism experiments, *Nat. Photo.* **2015**, *9*, 93.
- (13) Hao, C.; Xu, L.; Ma, W.; Wu, X.; Wang, L.; Kuang, H.; Xu, C., Unusual

Circularly Polarized Photocatalytic Activity in Nanogapped Gold-Silver Chiroplasmonic Nanostructures, *Adv. Funct. Mater.* **2015**, *25*, 5816.

(14) Kim, Y.; Yeom, B.; Arteaga, O.; Yoo, S. J.; Lee, S. G.; Kim, J. G.; Kotov, N. A., Reconfigurable chiroptical nanocomposites with chirality transfer from the macro- to the nanoscale, *Nat. Mater.* **2016**, *15*, 461.

(15) Feng, W.; Kim, J. Y.; Wang, X.; Calcaterra, H. A.; Qu, Z.; Meshi, L.; Kotov, N. A., Assembly of mesoscale helices with near-unity enantiomeric excess and light-matter interactions for chiral semiconductors, *Science advances* **2017**, *3*, e1601159.

(16) Giner-Casares, J. I.; Henriksen-Lacey, M.; Garcia, I.; Liz-Marzan, L. M., Plasmonic Surfaces for Cell Growth and Retrieval Triggered by Near-Infrared Light, *Angew. Chem. Int. Ed.* **2016**, *55*, 974.

(17) Li, W.; Chen, Z. W.; Zhou, L.; Li, Z. H.; Ren, J. S.; Qu, X. G., Noninvasive and Reversible Cell Adhesion and Detachment via Single-Wavelength Near-Infrared Laser Mediated Photoisomerization, *J. Am. Chem. Soc.* **2015**, *137*, 8199.

(18) Webber, M. J.; Appel, E. A.; Meijer, E. W.; Langer, R., Supramolecular biomaterials, *Nat. Mater.* **2016**, *15*, 13.

(19) Dong, R.; Zhou, Y.; Huang, X.; Zhu, X.; Lu, Y.; Shen, J., Functional Supramolecular Polymers for Biomedical Applications, *Adv. Mater.* **2015**, *27*, 498.

(20) Marchesan, S.; Styan, K. E.; Easton, C. D.; Waddington, L.; Vargiu, A. V., Higher and lower supramolecular orders for the design of self-assembled heterochiral tripeptide hydrogel biomaterials, *J. Mater. Chem. B* **2015**, *3*, 8123.

(21) Marchesan, S.; Easton, C. D.; Styan, K. E.; Waddington, L. J.; Kushkaki, F.; Goodall, L.; McLean, K. M.; Forsythe, J. S.; Hartley, P. G., Chirality effects at each amino acid position on tripeptide self-assembly into hydrogel biomaterials, *Nanoscale* **2014**, *6*, 5172.

(22) Solanki, A.; Chueng, S. T. D.; Yin, P. T.; Kappera, R.; Chhowalla, M.; Lee, K.-B., Axonal Alignment and Enhanced Neuronal Differentiation of Neural Stem Cells on Graphene-Nanoparticle Hybrid Structures, *Adv. Mater.* **2013**, *25*, 5477.

(23) Kang, K.; Park, Y. S.; Park, M.; Jang, M. J.; Kim, S. M.; Lee, J.; Choi, J. Y.; Jung, D. H.; Chang, Y. T.; Yoon, M. H., Axon-First Neuritegenesis on Vertical Nanowires,

Nano Lett. **2015**, 16, 675.

(24)Cho, W. K.; Kang, K.; Kang, G.; Jang, M. J.; Nam, Y.; Choi, I. S., Pitch-Dependent Acceleration of Neurite Outgrowth on Nanostructured Anodized Aluminum Oxide Substrates, *Angew. Chem. Int. Ed.* **2010**, 49, 10114.

(25)Kang, K.; Choi, S. E.; Jang, H. S.; Cho, W. K.; Nam, Y.; Choi, I. S.; Lee, J. S., In Vitro Developmental Acceleration of Hippocampal Neurons on Nanostructures of Self-Assembled Silica Beads in Filopodium-Size Ranges, *Angew. Chem. Int. Ed.* **2012**, 51, 2855.

(26)Zhang, R.; Li, Y.; Hu, B.; Lu, Z.; Zhang, J.; Zhang, X., Traceable Nanoparticle Delivery of Small Interfering RNA and Retinoic Acid with Temporally Release Ability to Control Neural Stem Cell Differentiation for Alzheimer's Disease Therapy, *Adv. Mater.* **2016**, 28, 6345.

(27)Critchley, K.; Khanal, B. P.; Gorzny, M. L.; Vigderman, L.; Evans, S. D.; Zubarev, E. R.; Kotov, N. A., Near-Bulk Conductivity of Gold Nanowires as Nanoscale Interconnects and the Role of Atomically Smooth Interface, *Adv. Mater.* **2010**, 22, 2338.

(28)Tang, M.; Song, Q.; Li, N.; Jiang, Z.; Huang, R.; Cheng, G., Enhancement of electrical signaling in neural networks on graphene films, *Biomaterials* **2013**, 34, 6402.

(29)Akhavan, O.; Ghaderi, E., Differentiation of human neural stem cells into neural networks on graphene nanogrids, *J. Mater. Chem. B* **2013**, 1, 6291.

(30)Lovat, V.; Pantarotto, D.; Lagostena, L.; Cacciari, B.; Grandolfo, M.; Righi, M.; Spalluto, G.; Prato, M.; Ballerini, L., Carbon nanotube substrates boost neuronal electrical signaling, *Nano Lett.* **2005**, 5, 1107.

(31)Martinelli, V.; Cellot, G.; Toma, F. M.; Long, C. S.; Caldwell, J. H.; Zentilin, L.; Giacca, M.; Turco, A.; Prato, M.; Ballerini, L.; Mestroni, L., Carbon Nanotubes Promote Growth and Spontaneous Electrical Activity in Cultured Cardiac Myocytes, *Nano Lett.* **2012**, 12, 1831.

(32)Jakus, A. E.; Secor, E. B.; Rutz, A. L.; Jordan, S. W.; Hersam, M. C.; Shah, R. N., Three-Dimensional Printing of High-Content Graphene Scaffolds for Electronic and

Biomedical Applications, *ACS Nano* **2015**, *9*, 4636.

(33)Guo, W.; Zhang, X.; Yu, X.; Wang, S.; Qiu, J.; Tang, W.; Li, L.; Liu, H.; Wang, Z. L., Self-Powered Electrical Stimulation for Enhancing Neural Differentiation of Mesenchymal Stem Cells on Graphene-Poly(3,4-ethylenedioxythiophene) Hybrid Microfibers, *ACS Nano* **2016**, *10*, 5086.

(34)Zhang, H.; Shih, J.; Zhu, J.; Kotov, N. A., Layered Nanocomposites from Gold Nanoparticles for Neural Prosthetic Devices, *Nano Lett.* **2012**, *12*, 3391.

(35)Liu, Y.; Wang, Y.; Claus, R. O., Layer-by-layer ionic self-assembly of Au colloids into multilayer thin-films with bulk metal conductivity, *Chem. Phys. Lett.* **1998**, *298*, 315.

(36)Tee, Y. H.; Shemesh, T.; Thiagarajan, V.; Hariadi, R. F.; Anderson, K. L.; Page, C.; Volkmann, N.; Hanein, D.; Sivaramakrishnan, S.; Kozlov, M. M., Cellular chirality arising from the self-organization of the actin cytoskeleton, *Nat. Cell Biol.* **2015**, *17*, 445.

(37)Hein, J. E.; Blackmond, D. G., On the Origin of Single Chirality of Amino Acids and Sugars in Biogenesis, *Accounts Chem. Res.* **2012**, *45*, 2045.

(38)Adhikari, B.; Nanda, J.; Banerjee, A., Multicomponent hydrogels from enantiomeric amino acid derivatives: helical nanofibers, handedness and self-sorting, *Soft Matter* **2011**, *7*, 8913.

(39)Roche, C.; Sun, H.-J.; Prendergast, M. E.; Leowanawat, P.; Partridge, B. E.; Heiney, P. A.; Araoka, F.; Graf, R.; Spiess, H. W.; Zeng, X.; Ungar, G.; Percec, V., Homochiral Columns Constructed by Chiral Self-Sorting During Supramolecular Helical Organization of Hat-Shaped Molecules, *J. Am. Chem. Soc.* **2014**, *136*, 7169.

(40)Yao, X.; Hu, Y.; Cao, B.; Peng, R.; Ding, J., Effects of surface molecular chirality on adhesion and differentiation of stem cells, *Biomaterials* **2013**, *34*, 9001.

(41)El-Gindi, J.; Benson, K.; De Cola, L.; Galla, H. J.; Kehr, N. S., Cell Adhesion Behavior on Enantiomerically Functionalized Zeolite L Monolayers, *Angew. Chem. Int. Ed.* **2012**, *51*, 3716.

(42)Orza, A.; Soritau, O.; Olenic, L.; Diudea, M.; Florea, A.; Ciuca, D. R.; Mihiu, C.; Casciano, D.; Biris, A. S., Electrically Conductive Gold-Coated Collagen Nanofibers

for Placental-Derived Mesenchymal Stem Cells Enhanced Differentiation and Proliferation, *ACS Nano* **2011**, *5*, 4490.

(43) Xie, J.; MacEwan, M. R.; Li, X.; Sakiyama-Elbert, S. E.; Xia, Y., Neurite Outgrowth on Nanofiber Scaffolds with Different Orders, Structures, and Surface Properties, *ACS Nano* **2009**, *3*, 1151.

(44) Feng, Z. Q.; Wang, T.; Zhao, B.; Li, J.; Jin, L., Soft Graphene Nanofibers Designed for the Acceleration of Nerve Growth and Development, *Adv. Mater.* **2015**, *27*, 6462.

(45) Khurana, S.; Schouteden, S.; Manesia, J. K.; Santamaria-Martinez, A.; Huelsken, J.; Lacy-Hulbert, A.; Verfaillie, C. M., Outside-in integrin signalling regulates haematopoietic stem cell function via Periostin-Itgav axis, *Nat. Comm.* **2016**, *7*:13500

Reviewers' comments:

Reviewer #1 (Remarks to the Author):

The authors have well responded to the referee's report. The revised version of the manuscript is acceptable for publication.

Reviewer #2 (Remarks to the Author):

Authors revised their manuscript seriously and properly according to Referees' comments.

Their responses to my and other referees comments show now clearly the innovative part of their work in comparison to the other reported studies in this field.

Therefore now I strongly recommend the publication of this study in Nature Communications.

Reviewer #3 (Remarks to the Author):

I thank the authors for providing detailed answers to my questions about their study. I thought the 4-point statements of significance were very useful. I thought the summary of unique innovations of the paper was excellent and helped re-clarified its overall significance. I did find it slightly difficult understand the originality of the study. Notwithstanding, the interest I had in the technique for cell manipulation and retrieval. The fine-control of cell neurite outgrowth patterns with the underlying was interesting. The film allowed absorbance of light at frequencies unlikely to detrimentally affect attached cells. I really appreciate the further cell-related experiments that have strengthened the practical utility of the film technology. It was good to see that the application of other types of chiral molecules could also produce the same affects as the original system. This kind of verstility is very impactful.

The validation of cell viability after retrieval is also a very significant experiment confirming that the topography, light treatment and detachment processes did not seriously damage the cells, which is a strong biological concern in such unusual and strong physical environments. I am convinced by the strong affect between the chiral films on surface associating cells. This is made more convincing by the experiments whereby adjustments are made and the cell respond to them in characteristic ways.

I am less convinced by the idea that the platforms are good enough to bring about regenerative nerve repair. Although, I do accept that this study is focused on the materials and technique driven aspects and that the cell validation experiments are preliminary. I think you have made a good case for explaining the mechanism behind the selection, attachment, differentiation and growth. Protein adsorption at the surface and following interactions between associating cells and the patterned surface are critical. It is reasonable to assume that that adsorbed serum proteins buffer the cells from the energy of the laser. I wonder what structural changes they undergo from laser treatment. No doubt that fibronectin secreted by cells is an important component of the conditioning layer of adsorbed proteins attached to the chiral film, but there are also other serum proteins (from media), which adsorb onto the film surface and adapt their structure to the chiral formation. I agree there is potential to use the platform for nerve repair, although the functionality of the nerve connections and electrical signal propagation needs to be confirmed, in part, by functional data.

Who will be interested in reading the paper, and why?

Materials scientists/ chemists, cell and tissue engineers because of the novel chiral design at the material surface and the interesting inductive effects on neuroblastoma cells for potential in nerve repair and the ability to separate them from the inductive platform.

What are the main claims of the paper and how significant are they?

The main claims (x4)

1. Monolayer chiral Au nanoparticle (NP) film with circular dichroism (CD) peaks at 550 nm and 775 nm was first fabricated for cell growth, differentiation and retrieval. [Significant]
2. It was found that L-Pen-NP film could accelerate cell proliferation while D- Pen-NP film exhibited opposite effect, which could be ascribed to stereospecific interaction between fibronectin and chiral NP film. [Not so significant]
3. The twisted spatial conformation of NP at nanoscale provided significant guidance in neurite outgrowth. Cells on L-Pen-NP film displaying dipolar differentiation, whereas multipolar neurite outgrowth occurred on D- Pen-NP film. [Significant]
4. Due to the broad plasmon absorbance band in visible and near-infrared region, circularly polarized light (808 nm laser) was conducted for high efficiency non-invasive cell harvest. [Not so significant]

Is the paper likely to be one of the five most significant papers published in the discipline this year?
No

How does the paper stand out from others in its field? I don't think it stands out from others rather it is comparable in standing for novelty with others.

Are the claims novel? If not, which published papers compromise novelty?
Yes the claims are novel except the differential effects of chirality on cells

Are the claims convincing? If not, what further evidence is needed?
Mainly, the materials, materials chemistry is very convincing the cell work less convincing.

Are there other experiments or work that would strengthen the paper further?
Yes, experiments to continue the cell culture timing, data on cell and tissue functionality etc.

How much would further work improve it, and how difficult would this be? Would it take a long time?
The cell work to form proper nerve tissue and test functionality should not take more than 2 months to complete.

Are the claims appropriately discussed in the context of previous literature?
Yes

If the manuscript is unacceptable, is the study sufficiently promising to encourage the authors to resubmit? Unacceptability hinges on the cell studies and Yes

If the manuscript is unacceptable but promising, what specific work is needed to make it acceptable?
More cell work to demonstrate its viability and potential to be clinically useful or to be useful in cell culture context at an experimental level.

Reviewer #4 (Remarks to the Author):

The revised manuscript has well responded some questions. However, there are still a few problems need to be clearly answered. Thus a major revision is needed.

1. It should be noted that the size of AuNP had been changed to 47 ± 5 nm, while it was 50 ± 5 nm in Figure S1 of the original version. However, Figure S1 in both of these two versions showed the same TEM picture. I am confused why there could be two different results.

2. AFM measurement gives the results (Figure S7) that most AuNPs were closely attached in the NP film because the average vertical distance looked like no more than 20 nm except some bigger distances were about 30-40 nm. The results suggested that AuNPs on the film were in a high density, which is different from the drawing showing in Figure 1B. By the way, similar AFM experiments should be performed on the glass substrates.

3. The authors had given more results to show the amount of L- or D-Pen on cell morphology. However, the error bars in Figure R24 and R25 were very large. From these results, there could not have got any significant difference. More repeated experiments are suggested.

4. Both Pen and proteins have N element. Considering the concentrations of Pen, there should be plenty of Pen conjugated on AuNPs, while Figure S44 showed much more proteins absorbed on NP film, which cannot be well understood. If the authors want to clearly show the differences, the amount of Pen and proteins should be measured directly but not just using XPS to detect their existence on the surfaces.

5. My last question, as comment 4-7, had not been well answered. It cannot be imagined that a layer of absorbed protein on the surface can protect cell from photothermal damage but detached them even the layer might be very thin and the environmental temperature is pretty high.

Response to Reviewers

Reviewers' comments:

Reviewer #1 (Remarks to the Author):

The authors have well responded to the referee's report. The revised version of the manuscript is acceptable for publication.

Reply: Thanks a lot for your unequivocal approval of our revision.

Reviewer #2 (Remarks to the Author):

Authors revised their manuscript seriously and properly according to Referees' comments.

Their responses to my and other referees comments show now clearly the innovative part of their work in comparison to the other reported studies in this field.

Therefore now I strongly recommend the publication of this study in Nature Communications.

Reply: Thank you very much for your unequivocal approval of our revision.

Reviewer #3 (Remarks to the Author):

I thank the authors for providing detailed answers to my questions about their study. I thought the 4-point statements of significance were very useful. I thought the summary of unique innovations of the paper was excellent and helped re-clarified its overall significance. I did find it slightly difficult understand the originality of the study. Notwithstanding, the interest I had in the technique for cell manipulation and

retrieval. The fine-control of cell neurite outgrowth patterns with the underlying was interesting. The film allowed absorbance of light at frequencies unlikely to detrimentally affect attached cells. I really appreciate the further cell-related experiments that have strengthened the practical utility of the film technology. It was good to see that the application of other types of chiral molecules could also produce the same affects as the original system. This kind of verstility is very impactful.

Reply: Thank you very much for approval and nice comments.

The validation of cell viability after retrieval is also a very significant experiment confirming that the topography, light treatment and detachment processes did not seriously damage the cells, which is a strong biological concern in such unusual and strong physical environments. I am convinced by the strong affect between the chiral films on surface associating cells. This is made more convincing by the experiments whereby adjustments are made and the cell respond to them in characteristic ways.

Reply: Many thanks for your productive comments.

I am less convinced by the idea that the platforms are good enough to bring about regenerative nerve repair. Although, I do accept that this study is focused on the materials and technique driven aspects and that the cell validation experiments are preliminary. I think you have made a good case for explaining the mechanism behind the selection, attachment, differentiation and growth. Protein adsorption at the surface and following interactions between associating cells and the patterned surface are

critical. It is reasonable to assume that that adsorbed serum proteins buffer the cells from the energy of the laser. I wonder what structural changes they undergo from laser treatment. No doubt that fibronectin secreted by cells is an important component of the conditioning layer of adsorbed proteins attached to the chiral film, but there are also other serum proteins (from media), which adsorb onto the film surface and adapt their structure to the chiral formation. I agree there is potential to use the platform for nerve repair, although the functionality of the nerve connections and electrical signal propagation needs to be confirmed, in part, by functional data.

Reply: Thanks for your comments. In this study, chiral nanoparticle films were designed to promote cell response in type and numbers, and easily retrieval the cells without damage. Looking to the future, this platform might provide opportunity to control the axonal alignment and growth of neural cell derived neurons for the development of more effective treatment for spinal cord injuries.

In this text, the chiral Au NPs film was fabricated with high optical properties as biomaterial suitable for neural interface. All experimental results displayed that our proposed L-penicillamine functionalized Au NP film exhibited excellent promotion for cell adhesion, proliferation (**Figure 3C**) and neurite outgrowth (**Figure 4**).

Compared to carbon-based materials, conductivity of macroscale composite materials made from Au NPs is higher.¹⁻² Some researchers reported and verified the high electrical conductivity of the films guarantee electrical coupling between the cells and substrate.^{1,3-7}

As for the differentiated cells, the CPL detached populations were stable and

retain their state. After cells cultured on chiral films (with addition of retinoic acid) differentiated, the cells were irradiated with LCP (for L-Pen-NP film) or RCP (for D-Pen-NP film) for 5 min. Then, the detached differentiated cells were transferred to normal tissue culture plate. 24 h later, the morphology of cells were characterized with confocal image and the expression of N-myc oncoprotein was confirmed by western blot. As shown in **Figure S55** and **Figure S56**, the detached cells populations were stable and retain their differentiated state well. These results indicated that the chiral film could be used as a platform to guide cell differentiation and have potentials for repair of.

In order to evaluate the stability and ability to differentiate of the detached cells, we detached cells from chiral films using near-infrared circularly polarized light and transferred them to another L- and D-Pen-NP film with addition of retinoic acid. The cells differentiated well after 6 days as shown in **Figure S45-46**. We elongated the differentiation time (8 days) of the detached cells and got high differentiated cells as **Figure R1**. The tight connections between neurites indicated that the detached cells remain the strong viability to differentiate and have good potentials for nerve damages repairing.

Is there any possibility for the structural changes of adsorbed serum proteins undergo laser treatment? No.

We analyzed the surface temperature of L-Pen-NP film under LCP illumination. As shown in **Figure R2**, the highest temperature (central temperature) of the film surface was 45.7 ± 0.1 °C. Previously study demonstrated that denaturation value for

secondary structure of bovine serum albumin was 50 °C⁸⁻¹¹. This revealed that the temperature condition in our research was safe for protein secondary structures.

Overall, this developed technique could facilitate the cell culture and cell engineering, including in the therapeutic use of cell lines.

Figure S55. Confocal images of the differentiated cells transferred to normal tissue culture plate 24 h later (A, cells differentiated on L-Pen-NP film and then were transferred to normal tissue culture plate. B, cells differentiated on D-Pen-NP film and then were transferred to normal tissue culture plate.) (red, actin; green, vinculin; blue, nucleus). Scale bars were 20 μ m.

Figure S56. Expression of N-myc protein in the differentiated cells and then transferred to normal tissue culture plate 24 h later (A, control. B, cells differentiated on D-Pen-NP film and then were transferred to normal tissue culture plate. C, cells differentiated on L-Pen-NP film and then were transferred to normal tissue culture plate.).

Figure S45. MTT assay of cells detached by NIR light and transferred to another L- (A) and (B) D-Pen-NP film.

Figure S46. Confocal images of cells detached by NIR light and transferred to another L- (A) and (B) D-Pen-NP film then differentiated (with addition of retinoic acid, 6 days). Red, actin; green, vinculin; blue, nucleus. Scale bars were 20 μ m.

Figure R1. Confocal images of cells detached by NIR light and transferred to another L- (A) and (B) D-Pen-NP film then differentiated (with addition of retinoic acid, 8 days). Red, actin; green, vinculin; blue, nucleus. Scale bars were 20 μ m.

Figure R2 (Figure S59). The thermal images of L-Pen-NP film under the irradiation of LCP for 5 min (808nm laser, 150 mW/cm²).

Who will be interested in reading the paper, and why?

Materials scientists/ chemists, cell and tissue engineers because of the novel chiral design at the material surface and the interesting inductive effects on neuroblastoma cells for potential in nerve repair and the ability to separate them from the inductive platform.

What are the main claims of the paper and how significant are they?

The main claims (x4)

1. Monolayer chiral Au nanoparticle (NP) film with circular dichroism (CD) peaks at 550 nm and 775 nm was first fabricated for cell growth, differentiation and retrieval. [Significant]

2. It was found that L-Pen-NP film could accelerate cell proliferation while D-Pen-NP film exhibited opposite effect, which could be ascribed to stereospecific

interaction between fibronectin and chiral NP film. [Not so significant]

3. The twisted spatial conformation of NP at nanoscale provided significant guidance in neurite outgrowth. Cells on L-Pen-NP film displaying dipolar differentiation, whereas multipolar neurite outgrowth occurred on D- Pen-NP film. [Significant]

4. Due to the broad plasmon absorbance band in visible and near-infrared region, circularly polarized light (808 nm laser) was conducted for high efficiency non-invasive cell harvest. [Not so significant]

Is the paper likely to be one of the five most significant papers published in the discipline this year? No

How does the paper stand out from others in its field? I don't think it stands out from others rather it is comparable in standing for novelty with others.

Are the claims novel? If not, which published papers compromise novelty?

Yes the claims are novel except the differential effects of chirality on cells

Are the claims convincing? If not, what further evidence is needed?

Mainly, the materials, materials chemistry is very convincing the cell work less convincing.

Are there other experiments or work that would strengthen the paper further?

Yes, experiments to continue the cell culture timing, data on cell and tissue functionality etc.

How much would further work improve it, and how difficult would this be? Would it take a long time?

The cell work to form proper nerve tissue and test functionality should not take more than 2 months to complete.

Are the claims appropriately discussed in the context of previous literature?

Yes

If the manuscript is unacceptable, is the study sufficiently promising to encourage the authors to resubmit? Unacceptability hinges on the cell studies and Yes

If the manuscript is unacceptable but promising, what specific work is needed to make it acceptable? More cell work to demonstrate its viability and potential to be clinically useful or to be useful in cell culture context at an experimental level.

Reply: Thank you very much for constructive and approval comments.

Reviewer #4 (Remarks to the Author):

The revised manuscript has well responded some questions. However, there are still a few problems need to be clearly answered. Thus a major revision is needed.

1. It should be noted that the size of AuNP had been changed to 47 ± 5 nm, while it was 50 ± 5 nm in Figure S1 of the original version. However, Figure S1 in both of these two versions showed the same TEM picture. I am confused why there could be two different results.

Reply: Thanks for your nice comments. We are very sorry about that. The Au nanoparticles used in this study were at the size of 47 ± 5 nm. The diameter '50 \pm 5 nm' in Figure S1 of the original version was not accurate, which was ignored in the manuscript preparation.

Many thanks for your kind reminder. All have been corrected.

2. AFM measurement gives the results (Figure S7) that most AuNPs were closely attached in the NP film because the average vertical distance looked like no more than 20 nm except some bigger distances were about 30-40 nm. The results suggested that AuNPs on the film were in a high density, which is different from the drawing showing in Figure 1B. By the way, similar AFM experiments should be performed on the glass substrates.

Reply: Thanks for your nice suggestions. According to the results of AFM and SEM, the Au NPs on the film displayed close packing, and we revised scheme 1B shown as

follows. Many thanks for your kind reminding.

Also, similar AFM experiment result of Au NP film on glass substrate has been done as shown in **Figure R3**. As we can see from the upper and lower end of the measuring area in z axis in **Figure R4**, the vertical dimension was also approximately 47 nm.

Figure 1B. Schematic presentation of the regulation of NG108-15 cell adhesion by stereospecific interaction between fibronectin and chiral NP film.

Figure R3 (Figure S57). AFM 3D image of Au NP film on glass substrate.

Figure R4 (Figure S58). Vertical distance of Au NP film on glass substrate.

3. The authors had given more results to show the amount of L- or D-Pen on cell morphology. However, the error bars in Figure R24 and R25 were very large. From these results, there could not have got any significant difference. More repeated experiments are suggested.

Reply: Thanks for your comments and suggestion. For the effects of the amount of L- or D-Pen on cell morphology, more repeated experiments have been done and the statistical results were exhibited in **Figure R5-R14**. As for cell adhesion and differentiation, there are some trends. The average aspect ratio of cells adhered on L-Pen-NP film increased with the increase of amount of L-Pen (**Figure R5-R6, R9**). On the contrary, cells adhered on D-Pen-NP film exhibited smaller sticking area when the amounts of D-Pen increased (**Figure R7-R8**).

And the average length of neurites increased with the increase of Pen amounts (**Figure R10-R14**) after RA was added.

Figure R5. Confocal images of NG108-15 cells adhered on L-Pen-NP films functionalized with 20 mM L-Pen (8 hr) without addition of retinoic acid. (Red, actin; green, vinculin; blue, nucleus). Scale bars were 20 μ m.

Figure R6. Confocal images of NG108-15 cells adhered on L-Pen-NP films functionalized with 40 mM L-Pen (8 hr) without addition of retinoic acid. (Red, actin; green, vinculin; blue, nucleus). Scale bars were 20 μ m.

Figure R7. Confocal images of NG108-15 cells adhered on D-Pen-NP films functionalized with 20 mM D-Pen (8 hr) without addition of retinoic acid. (Red, actin; green, vinculin; blue, nucleus). Scale bars were 20 μ m.

Figure R8. Confocal images of NG108-15 cells adhered on D-Pen-NP films functionalized with 40 mM D-Pen (8 hr) without addition of retinoic acid. (Red, actin; green, vinculin; blue, nucleus). Scale bars were 20 μ m.

Figure R9. Aspect ratio of NG 108-15 cells cultured on L- or D-Pen-NP film modified with different amounts of L/D-Pen without addition of RA (n = 6).

Figure R10. Confocal images of NG108-15 cells adhered on L-Pen-NP films functionalized with 20 mM L-Pen with addition of retinoic acid (RA, 6 days). (Red, actin; green, vinculin; blue, nucleus). Scale bars were 20µm.

Figure R11. Confocal images of NG108-15 cells adhered on L-Pen-NP films functionalized with 40 mM L-Pen with addition of retinoic acid (RA, 6 days). (Red, actin; green, vinculin; blue, nucleus). Scale bars were 20 μ m.

Figure R12. Confocal images of NG108-15 cells adhered on D-Pen-NP films functionalized with 20 mM D-Pen with addition of retinoic acid (RA, 6 days). (Red, actin; green, vinculin; blue, nucleus). Scale bars were 20 μ m.

Figure R13. Confocal images of NG108-15 cells adhered on D-Pen-NP films functionalized with 40 mM D-Pen with addition of retinoic acid (RA, 6 days). (Red, actin; green, vinculin; blue, nucleus). Scale bars were 20 μ m.

Figure R14. Mean lengths of neurites of NG108-15 cells differentiated on chiral films modified with different amounts of L/D-Pen with addition of RA (n = 6).

4. Both Pen and proteins have N element. Considering the concentrations of Pen, there should be plenty of Pen conjugated on AuNPs, while Figure S44 showed much more proteins absorbed on NP film, which cannot be well understood. If the authors want to clearly show the differences, the amount of Pen and proteins should be measured directly but not just using XPS to detect their existence on the surfaces.

Reply: Many thanks for your constructive comments. To measure the whole amount of Pen and proteins, prepared Au NP films were scraped off from the glass substrates carefully and suspended in 50% ethanol for test. Both Pen and proteins have N element, and we first measured Pen amounts by detecting S element without protein. From the spectra of Au and S (**Figure R15**), the L-Pen-NP film and D-Pen-NP film exhibited similar amounts of the S and Au elements, indicating the similar amounts of

L-Pen and D-Pen grafted onto the Au NP films.

According to the S element test results, there was plenty of Pens conjugated on AuNPs. However, proteins still could interact with the film surface (including the Au NP surface and Pen molecules).

Previously reports have demonstrated that brush polymer films based on amino acid has strong affinity with proteins.¹²⁻¹³ As the equal amounts of Pen modified to L/D-Pen-NP film, the N background of the films were equal too. Then, the Pen functionalized NP films were incubated in cell medium (10% fetal bovine serum) for 1 hour. To measure the amounts of adsorbed protein, N element was detected. As we can see from **Figure R16**, the N amount of L-Pen-NP film was higher than that of D-Pen-NP film, and the results displayed that L-Pen-NP film absorbed more proteins than D-Pen-NP film.

Figure R15. Amounts of (A) Au and (B) S element of the L/D-Pen (50 mM) modified Au NP films before protein adsorption.

Figure R16. N element of Pen (50 mM) modified Au NP films after protein adsorption.

The S element was tested by DIONEX ICS-3000. The Au element was tested by iCAP 6300. N element was acquired by Elementar vario EL cube.

5. My last question, as comment 4-7, had not been well answered. It cannot be imagined that a layer of absorbed protein on the surface can protect cell from photothermal damage but detached them even the layer might be very thin and the environmental temperature is pretty high.

Reply: Thanks for your comments. We are very sorry for not replying to this comment well.

“Why the other materials and conditions just reached a lower temperature after laser excitation but with a higher cell death”.

The reason why LCP light led to the high photothermal efficiency for L-Pen-NP film was that LCP light activated a larger number of L-Pen-NPs than other light

conditions (LP or RCP), which increased the number of photoejected hot electrons of chiral film by adsorbing the related rotation of polarized light.¹⁴

As we can see in **Figure R2**, the highest temperature of L-Pen-NP film under LCP was 45.7 ± 0.1 °C, which was lower than the generally agreed lethal value (47 °C)¹⁰⁻¹¹. In addition, cells almost detached from the film before 5 min and suspended in cell medium.

To be noticed, most of the previously reports used visible light or near infrared light, usually in the range of tens of W/cm².¹²⁻¹⁴ In our text, circularly polarized light (CPL) was applied for cell detachment with improved light efficacy for the first time. The values of power density (150 mW/cm², 5 min) are much lower than previous reports.¹⁵⁻¹⁷

The reasons why L-Pen-NP film protected the cells from photothermal damage were proved as follows:

Firstly, the protein adsorption experiment indicated that more proteins adsorbed on the L-Pen-NP film than D-Pen-NP film, which might work as a protecting layer (**Figure R16**). Moreover, we prepared two L-Pen-NP films with different treatments. One was incubated in cell medium (10% fetal bovine serum, FBS) at 37 °C for 12 h. And the other was incubated in PBS at 37 °C for 12 h as control. Then, the equal number of cells was dropped on the two treated films for LCP illumination (150 mW/cm², 5 min) respectively. As shown in **Figure R17**, cells on film which adsorbed proteins exhibited better survival rate. MTT assay was also conducted. The cell

viability of cells on L-Pen-NP film with protein adsorption kept 93.2 ± 6.4 %, while the control group displayed 72.2 ± 5.1 % (**Figure R18**). These experiments demonstrated that the protein layer protected the cells from photothermal damage.

Secondly, previously reports demonstrated that irradiation time and power densities required for cell detachment were different for each cell line.¹⁸⁻¹⁹ Under the same power density, the irradiation time required for detachment was relative shorter for cells those with large average cell area.¹⁵ The localized photothermal effect lead to the highly efficient detachment of cells, simultaneously avoiding significant damages to cells.

As for NG108-15 cells, most adherent cells showed stretching when grown on the L-Pen-NP film, whereas the cells on the D-Pen-NP film had a predominantly round morphology (**Figure 3A**). Cells on L-Pen-NP film display a comparatively larger area than those on D-Pen-NP film. This indicated the irradiation time required for detachment was shorter for cells on L-Pen-NP film, which led to cells detachment before 5 minutes, and the cell damage was significantly avoided.

Figure R2 (Figure S59). The thermal images of L-Pen-NP film under the irradiation of LCP for 5 min. (808nm laser, 150 mW/cm²).

Figure R17 (Figure S60). Live/dead assay (green, live; red, dead) conducted on NG108-15 cells which were dropped on the two treated films for LCP illumination (808nm laser, 150 mW/cm², 5 min) respectively. A) L-Pen-NP film incubated in cell medium (10% fetal bovine serum, FBS) at 37 °C for 12 h, and B) L-Pen-NP film was incubated in PBS at 37 °C for 12 h as control. Scale bars were 50 nm.

Figure R18 (Figure S61). MTT assay conducted on NG108-15 cells which were dropped on the two treated films for LCP illumination (808nm laser, 150 mW/cm², 5 min) respectively. A) L-Pen-NP film incubated in cell medium (10% fetal bovine serum, FBS) at 37 °C for 12 h, and B) L-Pen-NP film was incubated in PBS at 37 °C for 12 h as control. Scale bars were 50 nm.

The sentences in red have been added in the main text.

The reason why LCP light led to the high photothermal efficiency for L-Pen-NP film was that LCP light activated a larger number of L-Pen-NPs, which increased the number of photoejected hot electrons of chiral film by adsorbing the related rotation of polarized light.⁶⁷

Besides, cells on L-Pen-NP film displayed a comparatively larger area than those on D-Pen-NP film. The localized photothermal effect led to the highly efficient detachment of cells, simultaneously avoiding significant damages to cells.

References

- (1) Zhang, H.; Shih, J.; Zhu, J.; Kotov, N. A., Layered Nanocomposites from Gold Nanoparticles for Neural Prosthetic Devices, *Nano Letters* **2012**, *12*, 3391.
- (2) Liu, Y.; Wang, Y.; Claus, R. O., Layer-by-layer ionic self-assembly of Au colloids into multilayer thin-films with bulk metal conductivity, *Chemical Physics Letters* **1998**, *298*, 315.
- (3) Tang, M.; Song, Q.; Li, N.; Jiang, Z.; Huang, R.; Cheng, G., Enhancement of electrical signaling in neural networks on graphene films, *Biomaterials* **2013**, *34*, 6402.
- (4) Lovat, V.; Pantarotto, D.; Lagostena, L.; Cacciari, B.; Grandolfo, M.; Righi, M.; Spalluto, G.; Prato, M.; Ballerini, L., Carbon nanotube substrates boost neuronal electrical signaling, *Nano Letters* **2005**, *5*, 1107.
- (5) Martinelli, V.; Cellot, G.; Toma, F. M.; Long, C. S.; Caldwell, J. H.; Zentilin, L.; Giacca, M.; Turco, A.; Prato, M.; Ballerini, L.; Mestroni, L., Carbon Nanotubes Promote Growth and Spontaneous Electrical Activity in Cultured Cardiac Myocytes, *Nano Letters* **2012**, *12*, 1831.
- (6) Akhavan, O.; Ghaderi, E., Differentiation of human neural stem cells into neural networks on graphene nanogrids, *Journal of Materials Chemistry B* **2013**, *1*, 6291.
- (7) Jakus, A. E.; Secor, E. B.; Rutz, A. L.; Jordan, S. W.; Hersam, M. C.; Shah, R. N., Three-Dimensional Printing of High-Content Graphene Scaffolds for Electronic and Biomedical Applications, *Acs Nano* **2015**, *9*, 4636.
- (8) Militello, V.; Vetri, V.; Leone, M., Conformational changes involved in thermal aggregation processes of bovine serum albumin, *Biophysical Chemistry* **2003**, *105*, 133.
- (9) Moriyama, Y.; Watanabe, E.; Kobayashi, K.; Harano, H.; Inui, E.; Takeda, K., Secondary Structural Change of Bovine Serum Albumin in Thermal Denaturation up to 130 °C and Protective Effect of Sodium Dodecyl Sulfate on the Change, *The Journal of Physical Chemistry B* **2008**, *112*, 16585.
- (10) Stephenson, N. G., Effects of temperature on reptilian and other cells, *Journal of embryology and experimental morphology* **1966**, *16*, 455.

- (11) Verdeny, I.; Fontaine, A.; Farre, A.; Montesusategui, M.; Martinbadosa, E., Heating effects on NG108 cells induced by laser trapping, *Proceedings of SPIE* **2011**, 809724.
- (12) Wang, X.; Gan, H.; Sun, T., Chiral Design for Polymeric Biointerface: The Influence of Surface Chirality on Protein Adsorption, *Advanced Functional Materials* **2011**, *21*, 3276.
- (13) Zhou, F.; Yuan, L.; Li, D.; Huang, H.; Sun, T.; Chen, H., Cell adhesion on chiral surface: The role of protein adsorption, *Colloids and Surfaces B-Biointerfaces* **2012**, *90*, 97.
- (14) Hao, C.; Xu, L.; Ma, W.; Wu, X.; Wang, L.; Kuang, H.; Xu, C., Unusual Circularly Polarized Photocatalytic Activity in Nanogapped Gold-Silver Chiroplasmonic Nanostructures, *Advanced Functional Materials* **2015**, *25*, 5816.
- (15) Giner-Casares, J. I.; Henriksen-Lacey, M.; Garcia, I.; Liz-Marzan, L. M., Plasmonic Surfaces for Cell Growth and Retrieval Triggered by Near-Infrared Light, *Angewandte Chemie-International Edition* **2016**, *55*, 974.
- (16) Pallavicini, P.; Basile, S.; Chirico, G.; Dacarro, G.; D'Alfonso, L.; Dona, A.; Patrini, M.; Falqui, A.; Sironi, L.; Taglietti, A., Monolayers of gold nanostars with two near-IR LSPRs capable of additive photothermal response, *Chemical Communications* **2015**, *51*, 12928.
- (17) Li, W.; Chen, Z. W.; Zhou, L.; Li, Z. H.; Ren, J. S.; Qu, X. G., Noninvasive and Reversible Cell Adhesion and Detachment via Single-Wavelength Near-Infrared Laser Mediated Photoisomerization, *Journal of the American Chemical Society* **2015**, *137*, 8199.
- (18) Kolesnikova, T. A.; Kohler, D.; Skirtach, A. G.; Mohwaldt, H., Laser-Induced Cell Detachment, Patterning, and Regrowth on Gold Nanoparticle Functionalized Surfaces, *Acs Nano* **2012**, *6*, 9585.
- (19) You, J.; Heo, J. S.; Kim, J.; Park, T.; Kim, B.; Kim, H.-S.; Choi, Y.; Kim, H. O.; Kim, E., Noninvasive Photodetachment of Stem Cells on Tunable Conductive Polymer Nano Thin Films: Selective Harvesting and Preserved Differentiation Capacity, *Acs Nano* **2013**, *7*, 4119.

Reviewers' comments:

Reviewer #1 (Remarks to the Author):

In my opinion, the authors have well responded to the reports of all of the referees. Although some referees remain some questions, there are no critical questions which could prohibit the publication of this manuscript. Anyway, the lines of all of the referees are helpful, and the quality of the manuscript has been improved after the two rounds of revision.

I suggest the publication of the revised version of the manuscript.

Reviewer #4 (Remarks to the Author):

The revised manuscript has well responded most of my questions. However, there are still a few problems need to be clearly answered. Thus some revisions are needed.

1. The statistic analysis of the differences in Figure R9 and R14 should be provided.
2. Both Pen and protein not only have just N element, but also S element. If the authors want to use these elements to explain the difference in protein adsorption, these elements must be measured in Figure R15 and R16. And in these experiments unmodified film has to be added as a control. Moreover, there are lots of methods can be used to detect protein adsorption, such as isotope-labeling and fluorescence-labeling.
3. Although the maximum vertical distance in Figure R4 showed as around 47 nm, it should be noticed that the distance is measured at the horizontal distance of about 1500 nm. Since the experiment was done on glass substrate, it cannot be promised the smoothness of the substrate and no NP layers formation.
4. Ref 10 in the Response (Stephenson, N. G., Effects of temperature on reptilian and other cells, Journal of embryology and experimental morphology 1966, 16, 455) doesn't support animal cell growth at above 45C. By the way, I cannot get Ref 11 for the evaluation.

Response to Reviewers

Reviewer #1 (Remarks to the Author):

In my opinion, the authors have well responded to the reports of all of the referees. Although some referees remain some questions, there are no critical questions which could prohibit the publication of this manuscript. Anyway, the lines of all of the referees are helpful, and the quality of the manuscript has been improved after the two rounds of revision.

I suggest the publication of the revised version of the manuscript.

Reply: Thank you very much for your unequivocal approval of our revision.

Reviewer #4 (Remarks to the Author):

The revised manuscript has well responded most of my questions. However, there are still a few problems need to be clearly answered. Thus some revisions are needed.

1. The statistic analysis of the differences in Figure R9 and R14 should be provided.

Reply: Many thanks for your nice suggestion. For the effects of the amount of L- or D-Pen on cell morphology (**Figure R1**), the average aspect ratio of cells adhered on L-Pen-NP film increased with the increase of amount of L-Pen. Cells on L-Pen(20 mM)-NP film exhibited an average aspect ratio of 2.7 ± 0.3 , which was lower than that on L-Pen(40 mM, 50 mM)-NP film (3.4 ± 0.5 , 5.3 ± 1.2 respectively).

As for cells adhered on D-Pen-NP films, although there was no obvious difference between the average aspect ratio of cells (about 1.5), the cell sticking area displayed smaller when the amounts of D-Pen increased (**Figure R2-3**, 50 mM D-Pen functionalized Au NP film in main text). With addition of RA, the average length of neurites showed increasing trend with the increase of D-Pen amounts (**Figure R4**). In addition, we had added the statistic results and analysis in the revised version of Supporting Information.

Figure R1 (Figure S62, Figure R9 in last version). Aspect ratio of NG 108-15 cells cultured on L- or D-Pen-NP film modified with different amounts of L/D-Pen without addition of RA (n = 6).

Figure R2 (Figure S63). Confocal images of NG108-15 cells adhered on D-Pen-NP films functionalized with 20 mM D-Pen (8 hr) without addition of retinoic acid. (Red, actin; green, vinculin; blue, nucleus). Scale bars were 20 μ m.

Figure R3 (Figure S64). Confocal images of NG108-15 cells adhered on D-Pen-NP films functionalized with 40 mM D-Pen (8 hr) without addition of retinoic acid. (Red, actin; green, vinculin; blue, nucleus). Scale bars were 20 μ m.

Figure R4 (Figure S65, Figure R14 in last version). Mean lengths of neurites of NG108-15 cells differentiated on chiral films modified with different amounts of L/D-Pen with addition of RA (n = 6).

2. Both Pen and protein not only have just N element, but also S element. If the authors want to use these elements to explain the difference in protein adsorption, these elements must be measured in Figure R15 and R16. And in these experiments unmodified film has to be added as a control. Moreover, there are lots of methods can be used to detect protein adsorption, such as isotope-labeling and fluorescence-labeling.

Reply: Thanks for your nice suggestion.

We have added Au NP film (without Pen modification and without protein adsorption) as control. The Au element amount of Au NP film was $46.8 \pm 2.4 \mu\text{g}/\text{cm}^2$ (**Figure R5**), and the amounts of N and S element were lower than the limit of detection.

The Au, S, and N elements amounts of L/D-Pen (50 mM) modified Au NP films were shown in **Figure R6** (L-Pen-NP film: Au, $47.2 \pm 2.5 \mu\text{g}/\text{cm}^2$; S, $0.16 \pm 0.01 \mu\text{g}/\text{cm}^2$; N, $0.07 \pm 0.01\%$. D-Pen-NP film: Au, $46.7 \pm 2.6 \mu\text{g}/\text{cm}^2$; S, $0.15 \pm 0.01 \mu\text{g}/\text{cm}^2$; N, $0.08 \pm 0.01\%$). The L-Pen-NP film and D-Pen-NP film exhibited similar amounts of

the Au, N and S elements, indicating the similar amounts of L-Pen and D-Pen grafted onto the Au NP films.

After the Pen functionalized NP films were incubated in cell medium (10% fetal bovine serum), Au, N and S elements were detected. The Au, S, and N elements amounts of L/D-Pen (50 mM) modified Au NP films were shown in **Figure R7** (L-Pen-NP film: Au, $46.9 \pm 2.4 \mu\text{g}/\text{cm}^2$; S, $0.27 \pm 0.01 \mu\text{g}/\text{cm}^2$; N, $0.96 \pm 0.03\%$. D-Pen-NP film: Au, $47.5 \pm 2.3 \mu\text{g}/\text{cm}^2$; S, $0.21 \pm 0.01 \mu\text{g}/\text{cm}^2$; N, $0.61 \pm 0.02 \%$). The N amount of L-Pen-NP film was higher than that of D-Pen-NP film. Also, the S amount of L-Pen-NP film was a little higher than that of D-Pen-NP film.

These results further demonstrated that L-Pen-NP film absorbed more proteins than D-Pen-NP film.

Many thanks for your suggestion of other methods for protein adsorption detection. However, due to the limited equipment, it is convenient for us to use the isotope-labeling detection method.

Figure R5 (Figure S66). Amounts of Au element of Au NP film (without Pen modification and before protein adsorption). While the amounts of N and S element were lower than the limit of detection.

Figure R6 (Figure S67). Amounts of (A) Au, (B) N and (C) S element of the L/D-Pen (50 mM) modified Au NP films before protein adsorption.

Figure R7 (Figure S68). Amounts of (A) Au, (B) N and (C) S element of the L/D-Pen (50 mM) modified Au NP films after protein adsorption.

To measure the Au, N, and S elements respectively, the related fabricated Au NP films were scraped off from the glass substrates carefully and suspended in 50% ethanol for test. The S element was tested by ion chromatography (DIONEX ICS-3000, USA)¹⁻². The Au element was tested by were determined by using an inductively coupled plasma optical emission spectrometer (ICP-OES, Thermo scientific iCAP 6300)³⁻⁴. N element was acquired by Elementar vario EL cube (Analysensysteme GmbH, Germany)⁵⁻⁶.

References:

- (1) Lee, C. Y.; Ho, K. L.; Lee, D. J.; Su, A.; Chang, J. S., Electricity harvest from nitrate/sulfide-containing wastewaters using microbial fuel cell with autotrophic denitrifier, *Pseudomonas sp. C27, international journal of hydrogen energy* **2012**, *37*, 15827.
- (2) Li, J.; Zhuang, G.; Huang, K.; Lin, Y.; Xu, C.; Yu, S., Characteristics and sources of air-borne particulate in Urumqi, China, the upstream area of Asia dust, *Atmospheric Environment* **2008**, *42*, 776.
- (3) Sun, J.; Yang, X., Gold nanoclusters–Cu²⁺ ensemble-based fluorescence turn-on and real-time assay for acetylcholinesterase activity and inhibitor screening, *Biosensors and Bioelectronics* **2015**, *74*, 177.
- (4) Sun, J.; Yang, F.; Yang, X., Synthesis of functionalized fluorescent gold nanoclusters for acid phosphatase sensing, *Nanoscale* **2015**, *7*, 16372.
- (5) Wang, K.; Li, Q.; Liu, B.; Cheng, B.; Ho, W.; Yu, J., Sulfur-doped g-C₃N₄ with enhanced photocatalytic CO₂-reduction performance, *Applied Catalysis B: Environmental* **2015**, *176*, 44.
- (6) Zhu, D.; Li, L.; Cai, J.; Jiang, M.; Qi, J.; Zhao, X., Nitrogen-doped porous carbons from bipyridine-based metal-organic frameworks: Electrocatalysis for oxygen reduction reaction and Pt-catalyst support for methanol electrooxidation, *Carbon* **2014**, *79*, 544.

3. Although the maximum vertical distance in Figure R4 showed as around 47 nm, it should be noticed that the distance is measured at the horizontal distance of about 1500 nm. Since the experiment was done on glass substrate, it cannot be promised the

smoothness of the substrate and no NP layers formation.

Reply: Thanks for your comments. Some additional experiments and analysis were provided to confirm our previous results.

Figure R8 was AFM 3D image (Figure R4 in last response version) of Au NP film on glass substrate after software (Nanoscope Analysis) processing, and the corresponding original image was displayed as **Figure R9**. In the Nanoscope Analysis (software for AFM), the command button “flatten” could deduct the contribution of substrate roughness. **Figure R10** was the 2D image of Au NP film on glass substrate after flattening. With the help of “section” command, we could see the corresponding vertical distance distribution in 2D images. As shown in **Figure R11**, the point **a** was in the little crack of Au NP film, and point **b** was on the surface of flat NP film, the vertical distance was approximately equal to the diameter of Au NPs. Also, there are several regions in white color, the height of which were higher than the most of region. As displayed in **Figure R12**, the vertical distance between point **c** and **d** was approximately 60 nm. And this was caused by the occasionally displacement between the adjacent particles. Moreover, we chose **ten** Au NPs to carry out “section” command (**Figure R13**), the horizontal distance between point **e** and **f** was approximately 470 nm.

In summary, the Au NP film on glass substrate was monolayer.

Figure R8 (Figure S57). AFM 3D image of Au NP film on glass substrate.

Figure R9 (Figure S69). Original 2D image of Au NP film on glass substrate before flattening.

Figure R10 (Figure S70). 2D image of Au NP film on glass substrate after flattening.

Figure R11 (Figure S71). A) 2D image of Au NP film on glass substrate after flattening. B) Corresponding vertical distance.

Figure R12 (Figure S72). A) 2D image of Au NP film on glass substrate after flattening. B) Corresponding vertical distance.

Figure R13 (Figure S73). A) 2D image of Au NP film on glass substrate after flattening. B) Corresponding vertical distance.

4. Ref 10 in the Response (Stephenson, N. G., Effects of temperature on reptilian and other cells, Journal of embryology and experimental morphology 1966, 16, 455) doesn't support animal cell growth at above 45°C By the way, I cannot get Ref 11 for the evaluation.

Reply: Thanks a lot for your nice comments and kindly reminding.

Besides reptilian cells, Ref. 10 also studied the behavior and survival of embryonic cells at high temperatures (37.5-45°C), and demonstrated different types of cells may have the different temperature ranges. In this version we delete this reference according to your suggestions.

We are sorry about the mistake in Ref. 11. The revised format of Ref. 11 is as follows⁷.

The PDF of Ref. 11 could be downloaded from <http://dx.doi.org/10.1117/12.893352>, and it was also attached as a reference No.7. The authors investigated laser heating effects on NG108 cells. At temperature of up to 47.5 °C, irreversible cell damage is only of 36.8 % according to the cell population experiments. This phenomenon was constructive to our researches.

In our study, as we can see in **Figure R14**, the highest temperature of L-Pen-NP film under LCP was 45.7 ± 0.1 °C. In addition, cells almost were detached from the film before 5 min and suspended in cell medium.

Figure R14 (Figure S59). The thermal images of L-Pen-NP film under the irradiation of LCP for 5 min. (808nm laser, 150 mW/cm²).

(7) Verdeny, I.; Fontaine, A.; Farre, A.; Montesusategui, M.; Martinbadosa, E., Heating effects

on NG108 cells induced by laser trapping, *Proceedings of Society of Photo-Optical Instrumentation Engineers* **2011**, 8097: 809724-1, doi: 10.1117/12.893352; <http://dx.doi.org/10.1117/12.893352>.

References:

- (1) Lee, C. Y.; Ho, K. L.; Lee, D. J.; Su, A.; Chang, J. S., Electricity harvest from nitrate/sulfide-containing wastewaters using microbial fuel cell with autotrophic denitrifier, *Pseudomonas* sp. C27, *international journal of hydrogen energy* **2012**, *37*, 15827.
- (2) Li, J.; Zhuang, G.; Huang, K.; Lin, Y.; Xu, C.; Yu, S., Characteristics and sources of air-borne particulate in Urumqi, China, the upstream area of Asia dust, *Atmospheric Environment* **2008**, *42*, 776.
- (3) Sun, J.; Yang, X., Gold nanoclusters–Cu²⁺ ensemble-based fluorescence turn-on and real-time assay for acetylcholinesterase activity and inhibitor screening, *Biosensors and Bioelectronics* **2015**, *74*, 177.
- (4) Sun, J.; Yang, F.; Yang, X., Synthesis of functionalized fluorescent gold nanoclusters for acid phosphatase sensing, *Nanoscale* **2015**, *7*, 16372.
- (5) Wang, K.; Li, Q.; Liu, B.; Cheng, B.; Ho, W.; Yu, J., Sulfur-doped g-C₃N₄ with enhanced photocatalytic CO₂-reduction performance, *Applied Catalysis B: Environmental* **2015**, *176*, 44.
- (6) Zhu, D.; Li, L.; Cai, J.; Jiang, M.; Qi, J.; Zhao, X., Nitrogen-doped porous carbons from bipyridine-based metal-organic frameworks: Electrocatalysis for oxygen reduction reaction and Pt-catalyst support for methanol electrooxidation, *Carbon* **2014**, *79*, 544.
- (7) Verdeny, I.; Fontaine, A.; Farre, A.; Montesusategui, M.; Martinbadosa, E., Heating effects on NG108 cells induced by laser trapping, *Proceedings of Society of Photo-Optical Instrumentation Engineers* **2011**, 8097: 809724-12, doi: 10.1117/12.893352; <http://dx.doi.org/10.1117/12.893352>.

REVIEWERS' COMMENTS:

Reviewer #4 (Remarks to the Author):

The authors have provided more results to answer the questions and most of the problems have been well responded. The revised manuscript is well written and can be accepted for publication.